# Object representations in the human brain reflect the co-occurrence statistics of vision and language

Michael F. Bonner[1,2✉] & Russell A. Epstein [2]

A central regularity of visual perception is the co-occurrence of objects in the natural environment. Here we use machine learning and fMRI to test the hypothesis that object co-occurrence statistics are encoded in the human visual system and elicited by the perception of individual objects. We identified low-dimensional representations that capture the latent statistical structure of object co-occurrence in real-world scenes, and we mapped these statistical representations onto voxel-wise fMRI responses during object viewing. We found that cortical responses to single objects were predicted by the statistical ensembles in which they typically occur, and that this link between objects and their visual contexts was made most strongly in parahippocampal cortex, overlapping with the anterior portion of scene-selective parahippocampal place area. In contrast, a language-based statistical model of the co-occurrence of object names in written text predicted responses in neighboring regions of object-selective visual cortex. Together, these findings show that the sensory coding of objects in the human brain reflects the latent statistics of object context in visual and linguistic experience.

[1] Department of Cognitive Science, Johns Hopkins University, Baltimore, MD, USA. [2] Department of Psychology, University of Pennsylvania, Philadelphia, PA, USA. ✉email: mfbonner@jhu.edu

Many objects have a natural place in the world—a setting in which they and other co-occurring objects are typically encountered. For example, fire hydrants are often found with traffic lights and mailboxes on city sidewalks, while tea kettles are found with mugs and stoves in kitchens. This type of contextual knowledge can help people identify their surroundings and generate expectations for the other objects they might encounter. Previous behavioral work has demonstrated that the visual system uses contextual knowledge to facilitate object recognition[1–3] and visual search[4]. Moreover, theoretical accounts suggest that contextual facilitation might be evidence for a general cognitive mechanism by which the mind makes predictions about the world in order to support adaptive behavior[5–10]. Thus, identifying the brain mechanisms that support the representation of contextual knowledge is an important challenge for cognitive neuroscience.

Previous neuroimaging studies have attempted to meet this challenge[11–14] One approach has been to operationalize context as a one-dimensional rating that reflects how strongly an object brings to mind a particular context[11,15] or whether or not an object is associated with other objects or locations[16]. Objects with strong contextual associations by these definitions were found to elicit greater responses than objects with weak contextual associations in the parahippocampal place area (PPA) and retrosplenial complex (RSComp), brain regions that are known to respond strongly during the visual perception of scenes[17,18]. Given that scenes are contexts by definition, typically containing several co-occurring objects, these results are consistent with the view that PPA and RSComp have a general role in both the perception and retrieval of contextual information[19]. However, because these conclusions were based on univariate analysis of regional responses along a single stimulus dimension, they were vulnerable to alternative explanations. Indeed, other object properties, such as real-world size and spatial stability, have been found to explain similar (or greater) variance in PPA and RSComp responses[20,21], and some of the effects of object context in scene-selective visual regions have not been consistently observed[22]. More importantly, this approach is limited insofar as it only examines whether an object has strong contextual associations or not; it does not attempt to analyze the nature of these associations (e.g., which objects are associated with each other). These limitations have led to uncertainty about whether scene regions play a central role in mediating contextual knowledge.

Here we attempt to resolve this issue by using an approach that more directly tests for contextual representations. Specifically, we use fMRI to search for neural representations that reflect the multivariate statistical structure of object co-occurrence in the visual environment. To model this statistical structure, we first developed an adaptation of the word2vec machine-learning algorithm from computational linguistics[23,24], which we label object2vec. Word2vec uses unsupervised learning to compress high-dimensional data about the co-occurrence of words within a large corpus of text into a relatively small number of highly informative dimensions. Object2vec performs the exact same operation for objects within a corpus of over 20,000 real-world scenes. Thus, object2vec creates a low-dimensional representational space that reflects the statistical regularities of object co-occurrence in natural images. We scanned human observers while they viewed isolated single objects, and we used voxel-wise encoding models to identify brain regions where the dimensions of object2vec predicted fMRI responses. Because object2vec is constructed to reflect the contextual associations of objects, we posited that if a brain region represents this contextual information, it should be well predicted by the object2vec embeddings. For comparison, we performed the same fMRI analysis on the dimensions of language-based word2vec.

Previous work has used related techniques from computational linguistics to examine the natural statistics of object co-occurrence and their possible neural correlates. Stansbury et al.[25] used a topic learning algorithm to define "categories" of object clusters that tend to co-occur within scenes, and then showed that voxel-wise fMRI response to scenes could be predicted based on their membership in these categories. This finding indicates that statistically defined scene categories may be an important factor underlying the organization of scene representations in the visual cortex. Our study builds on this approach to address a different set of questions about how the brain represents the contextual associations of individual objects, for which rich contextual information is not physically present in the visual stimulus. Sadeghi and colleagues[26] compared contextual information learned from object co-occurrence in images with contextual information learned from word co-occurrence in linguistic corpora. They found that while visual and linguistic contextual models are partially correlated, they also exhibit interesting patterns of divergence. Namely, visual context captures not only categorical similarities between objects (e.g., between items in the category of fruit) but also cross-category similarities that are not well accounted for by linguistic context (e.g., between fruit and kitchen items). Our study extends this modeling approach to determine how such visual and linguistic contextual information for objects is encoded in the brain.

To anticipate, our results show that parahippocampal cortex and the anterior portion of scene-selective PPA represent the statistical associations between objects and their visual contexts, and that these representations are elicited when subjects view individual objects that do not have any contextual information physically present in the stimulus. In contrast, language-based word2vec explained variance in neighboring object-selective regions (including posterior fusiform; pF), indicating that these regions represent object properties that are related to language-based co-occurrence statistics. Together these findings reveal the relationship between high-level object representations in the ventral visual cortex and the latent manifolds of object context in vision and language. Our results also provide insight into the statistical properties of cortical representations—specifically, they suggest that high-level sensory regions utilize an efficient coding scheme that compresses a large number of behaviorally relevant stimulus properties into a relatively small number of statistically informative dimensions.

## Results

**Object embeddings**. We first sought to characterize the co-occurrence statistics of objects in the visual environment. To accomplish this, we needed a large data set of natural images in which all object occurrences were labeled. We took advantage of the recently created ADE20K data set, which contains 22,210 annotated scenes in which every object has been manually labeled by an expert human annotator[27]. One approach for characterizing the co-occurrence statistics of this data set would be to simply construct a matrix of co-occurrence frequencies for all pairwise comparisons of objects. However, this would produce a sparse and high-dimensional matrix, with many redundant dimensions. In the field of computational linguistics, there is a long history of modeling word co-occurrence data in language corpora with dense, lower-dimensional representations[28]. This modeling framework, known as distributional semantics, has proved highly useful because the resulting representations capture information about the semantic properties of words, such that words with similar meanings tend to have similar representations. A leading technique for modeling distributional semantics is word2vec, which is a class of neural network models that are trained to

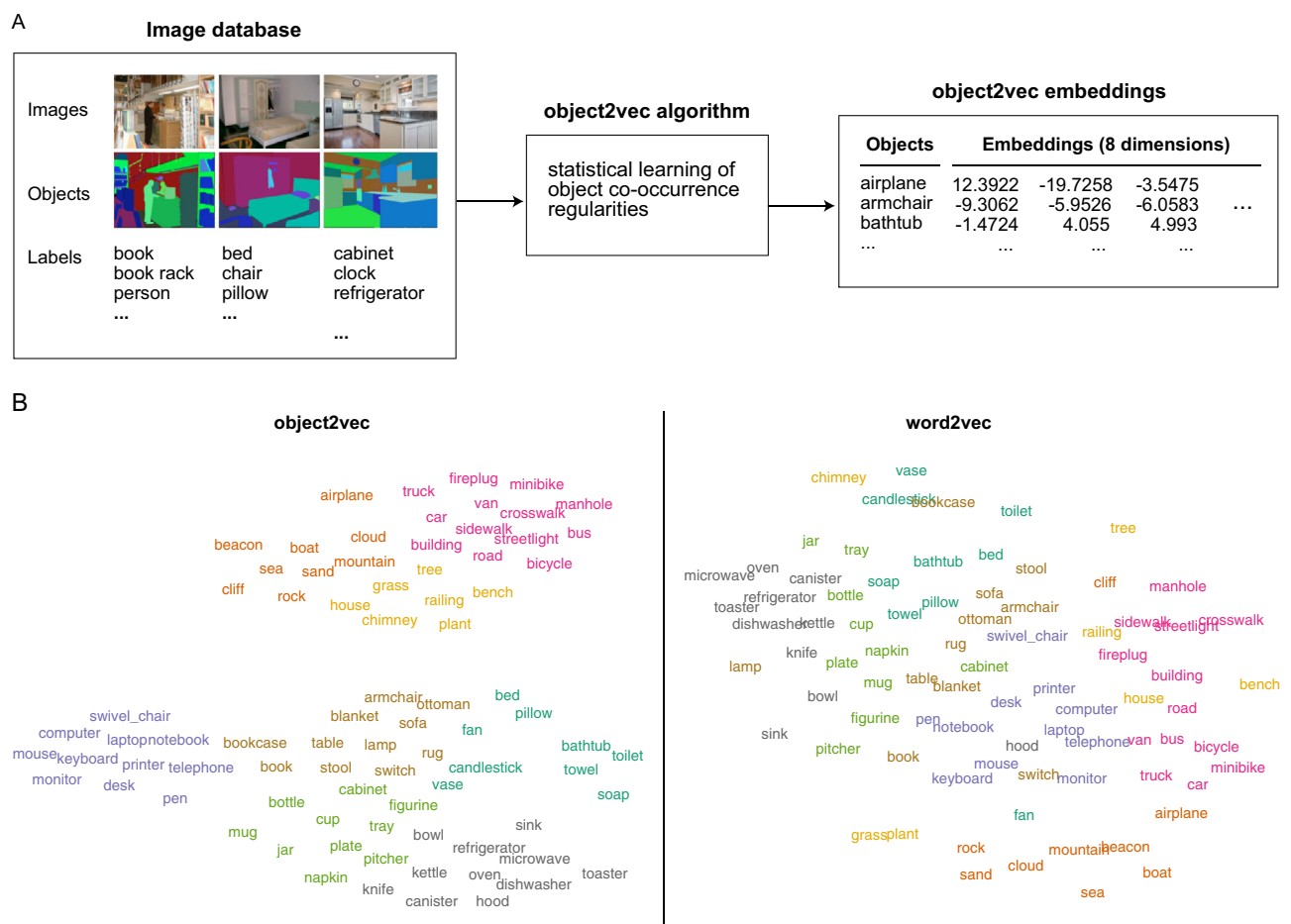

**Fig. 1 Statistical model of visual object context. A** Object context was modeled using the ADE20K data set[27], which contains 22,210 images in which every pixel is associated with an object label provided by an expert human annotator. An adaptation of the word2vec machine-learning algorithm for distributional semantics—which we call object2vec—was applied to this corpus of image annotations to model the statistical regularities of object-label co-occurrence in a large sample of real-world scenes. Using this approach, we generated a set of 8-dimensional object2vec representations for the object labels in the ADE20K data set. These representations capture the co-occurrence statistics of objects in the visual environment, such that objects that occur in similar scene contexts will have similar object2vec representations. **B** The panel on the left shows a two-dimensional visualization of the object2vec representations for the 81 object categories in the fMRI experiment. This visualization shows that the object2vec representations contain information about the scene contexts in which objects are typically encountered (e.g., indoor vs. outdoor, urban vs. natural, kitchen vs. office). The panel on the right shows a similar plot for the language-based word2vec representations of the same object categories. These plots were created using t-distributed stochastic neighbor embedding (tSNE). The colors reflect cluster assignments from k-means clustering and are included for illustration purposes only. K-means clustering was performed on the tSNE embeddings for object2vec, and the same cluster assignments were applied to the word2vec embeddings in the right panel for comparison. Source data are provided as a Source Data file.

predict target words from their surrounding linguistic contexts (or vice versa)[23,24]. The learned internal weights of word2vec models (known as word embeddings) reflect the latent statistical structure of the linguistic contexts in which words are typically found. Here we adopt the general approach of the word2vec algorithm to model the statistical structure of object context in images. We named this model object2vec.

In the object2vec model, object labels are treated like word tokens in a corpus and their contexts are defined by the images they occur in. Using this approach, we trained an object2vec model on the image annotations from ADE20K and learned a set of 8-dimensional embeddings that represent statistical information about the natural contexts of objects (Fig. 1A). Through a parameter search and principal component analysis (PCA), we found that higher-dimensional embeddings did not substantially alter the representational geometry of object2vec, suggesting that eight dimensions sufficed for learning representations of object context from this data set (see "Methods" section for details). Two-dimensional visualization of the object2vec embeddings (for the objects in our experiment) shows that these representations contain meaningful information about the way in which objects are grouped together in the natural environment, with object clusters that can be intuitively interpreted as familiar scene categories (e.g., indoor, outdoor, kitchen, office).

We also examined a language-based word2vec model. This model was trained on the co-occurrence of words in a natural language corpus containing ~100 billion words. Two-dimensional visualization of the word2vec representations suggests that they reflect a broader set of semantic associations than object2vec (Fig. 1B). These include information about the contexts in which objects are found, but also information about what kinds of taxonomic categories objects belong to, such as modes of transportation (e.g., airplane, car, bus, bicycle) and furniture for sitting on (e.g., swivel chair, armchair, sofa, stool).

**Cortical representations and object co-occurrence in vision and language.** With our models of vision- and language-based context in hand, we set out to test the hypothesis that object

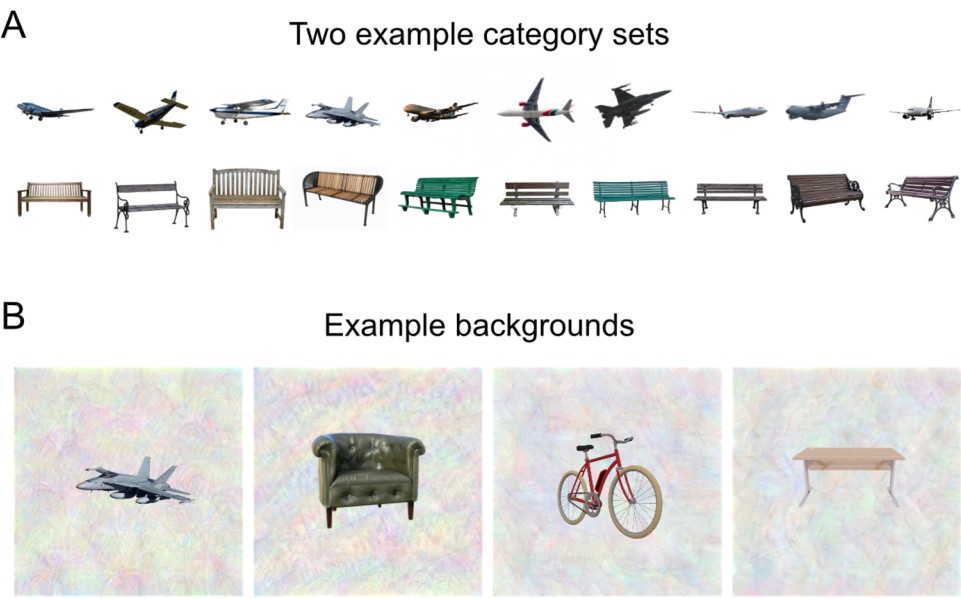

**Fig. 2 Experimental stimuli. A** In an fMRI experiment, subjects viewed images of isolated objects from 81 different categories. There were 10 unique tokens for each category (for a total of 810 unique stimuli). This panel shows 10 illustrative stimuli for two categories. **B** In the fMRI scanner, images were presented on complex, textured backgrounds to reduce the saliency of low-level features related to object shape and size. This panel shows four illustrative stimuli on textured backgrounds. All stimuli in this figure are shown for illustration purposes using images that we have the license to publish. They closely resemble the actual stimuli used in the fMRI experiment. Airplane image THY 9983 Nevit transparent by Nevit Dilmen is licensed under CC BY-SA 3.0.

representations in visual cortex reflect the statistical regularities captured by these models. We were specifically interested in determining whether such contextual representations are elicited whenever an object is viewed—even when subjects are not explicitly asked to report contextual information. Thus, it was important that our object stimuli be shown in isolation, without other co-occurring objects present in the image. We designed a stimulus set of 810 objects from 81 different categories (10 object tokens per category). Example stimuli are shown in Fig. 2. All object categories are listed in Supplementary Table 1, and all stimuli are available on the Open Science Framework repository for this project (see "Data availability" section). We measured fMRI responses while subjects viewed these objects, shown alone on meaningless textured backgrounds, and performed a simple perceptual task of responding by button press whenever they saw a "warped" object. Warped objects were created through diffeomorphic warping of object stimuli (Supplementary Fig. 1; see "Methods" section for details).

Using a voxel-wise modeling procedure, we examined whether fMRI responses to these object stimuli could be predicted from the co-occurrence statistics of objects in vision and language. We first estimated the fMRI responses to each object category. We then fit voxel-wise encoding models with the goal of predicting the fMRI responses to all object categories through a weighted linear sum of the object2vec or word2vec embeddings. Specifically, a set of linear regression models were estimated using the statistical embeddings as regressors and the voxel-wise fMRI responses as predictands (Fig. 3). Through cross-validation, we assessed how well the estimated encoding models could predict fMRI responses to out-of-sample stimuli (see "Methods" section for details). The cross-validation procedure was designed so that the training and test sets always contained objects from different categories, which allowed for a strong test of generalization to new semantic categories, rather than new stimuli from the same categories.

We first performed a region of interest (ROI) analysis of the PPA, which has been proposed to have a central role in the processing of object context[19,29]. We looked separately at the anterior and posterior PPA, because previous work suggests that non-spatial contextual associations elicit stronger responses in anterior parahippocampal cortex than posterior parahippocampal cortex[16] and because several lines of evidence suggest functional and connectivity differences across the anterior-posterior extent of the PPA[30,31]. Figure 4 shows results for both the image-based object2vec model and language-based word2vec (see Supplementary Fig. 2 for single-subject results). As can be seen in the ROI plot, both the split-half reliability of our data and the prediction accuracy of our encoding models are quite high for an fMRI experiment. We believe this is likely attributable to the fact that we used a mini-block design, with a large number of repeated presentations for each stimulus (although another factor may be our use of a reliability mask, which would bias the reliability scores upward; see "Methods" section). In terms of statistical significance, both object2vec and word2vec had significant prediction accuracy in both anterior and posterior segments of the PPA. Thus, at least some portion of the object-evoked responses throughout the PPA could be reliably predicted from representations of the visual and linguistic contexts of objects. However, in terms of the magnitude of the effects, the highest prediction accuracy was observed for the image-based statistics of the object2vec model in anterior PPA. In an interaction test, we found that the difference in accuracy between object2vec and word2vec was greater in anterior PPA compared to posterior PPA (permutation test, $p = 0.002$).

We next performed ROI analyses for other functionally defined regions, including the scene-selective RSComp, which has also been implicated in object context along with the PPA[11,29]. We also analyzed scene-selective occipital place area (OPA), PPA (without segmenting into anterior and posterior regions), object-selective pFs[32], object-selective lateral occipital (LO)[33], and early visual cortex (EVC). Results for these ROIs are shown in Fig. 5 (see Supplementary Fig. 3 for single-subject results). The image-based object2vec model generated significant prediction accuracies in all scene-selective ROIs and in one object-selective ROI (pFs). The language-based word2vec model generated significant prediction accuracies in all ROIs, with larger effects in the object-

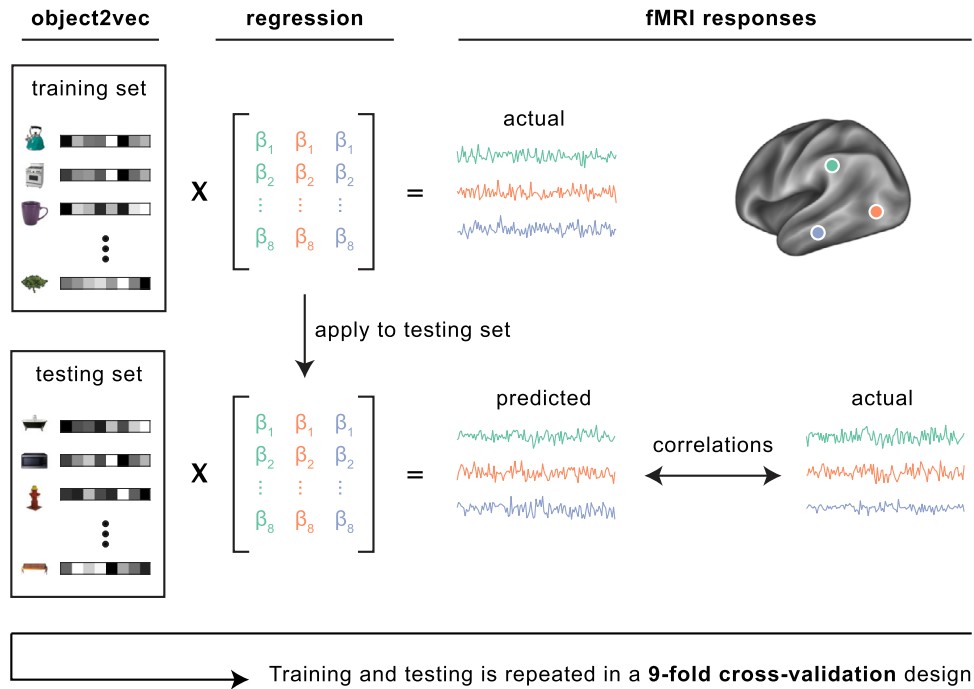

**Fig. 3 Encoding model of visual object context.** Voxel-wise encoding models were used to assess whether our statistical model of visual object context (object2vec) could reliably explain variance in the fMRI responses to the experimental object categories. Linear regression was used to map the representations of object2vec onto fMRI responses. We assessed the out-of-sample prediction accuracy of these regression models through a 9-fold cross-validation procedure. Each fold of the cross-validation design contains a set of object categories that do not appear in any other fold. These folds were shown in separate fMRI runs. Parameters for the voxel-wise linear regression models were estimated using the fMRI data for 8 folds of the object categories and the learned regression weights were then applied to the held-out object categories in the remaining fold to generate a set of predicted fMRI responses. This procedure was repeated for each fold of the cross-validation design, and the predicted fMRI responses from each fold were concatenated together. Prediction accuracy was assessed by calculating the voxel-wise correlations of the predicted and actual fMRI responses across all object categories.

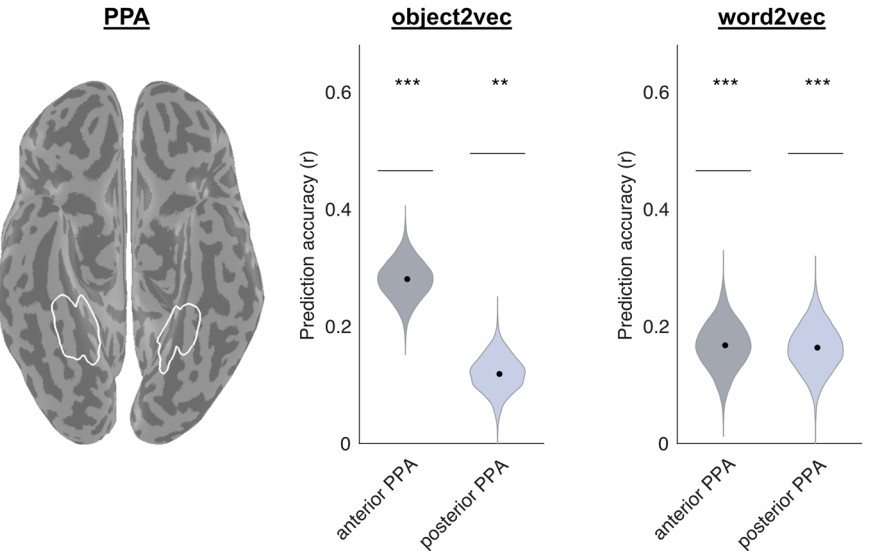

**Fig. 4 Encoding models of visual and linguistic context predict fMRI responses to objects in PPA.** This plot shows the average prediction accuracies for encoding models in voxels from the anterior third and posterior third of the PPA using either image-based object2vec representations as regressors or language-based word2vec representations. The violin plots show the mean prediction accuracies (central black dots) and bootstrap standard deviations. The gray lines above each violin plot indicate the average voxel-wise split-half reliability of the fMRI responses in each ROI. Prediction accuracies for both models were above chance, but the highest accuracy was observed for object2vec in the anterior PPA. The image on the left shows the PPA parcel on a ventral view of a cortical surface rendering. PPA parahippocampal place area. $**p < 0.01$, $***p < 0.001$, uncorrected, one-sided permutation test. Exact $p$-values for object2vec: anterior PPA $p$-value $= 2.0e-04$; posterior PPA $p$-value $= 2.4e-03$. Exact $p$-values for word2vec: anterior PPA $p$-value $= 4.0e-04$; posterior PPA $p$-value $= 2.0e-04$. Source data are provided as a Source Data file.

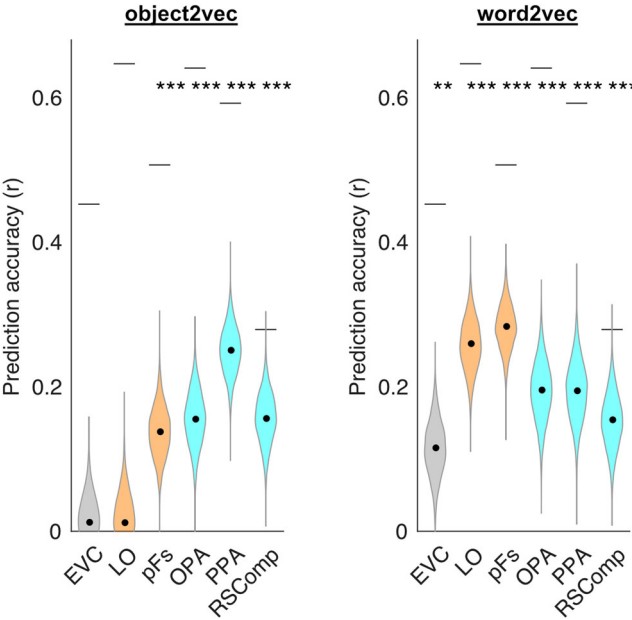

**Fig. 5 Encoding models of visual and linguistic context predict fMRI responses to objects in functionally defined ROIs.** This plot shows the average prediction accuracies for encoding models in voxels from multiple regions of interest using either image-based object2vec representations as regressors or language-based word2vec representations. Object-selective ROIs are plotted in orange and scene-selective ROIs are plotted in cyan. The violin plots show the mean prediction accuracies (central black dots) and bootstrap standard deviations. The gray lines above each violin plot indicate the average voxel-wise split-half reliability of the fMRI responses in each ROI. The highest prediction accuracy for object2vec was in the PPA, whereas the highest prediction accuracy for word2vec was in LO and pFS. EVC early visual cortex, LO lateral occipital, pFs posterior fusiform, OPA occipital place area, PPA parahippocampal place area, RSComp retrosplenial complex. **p < 0.01, ***p < 0.001, uncorrected, one-sided permutation test. Exact p-values for object2vec: EVC p-value = 3.8e−01; LO p-value = 4.0e−01; pFs p-value = 8.0e−04; OPA p-value = 4.0e-04; PPA p-value = 2.0e−04; RSComp p-value = 2.0e−04. Exact p-values for word2vec: EVC p-value = 2.4e−03; LO p-value = 2.0e−04; pFs p-value = 2.0e−04; OPA p-value = 2.0e−04; PPA p-value = 4.0e−04; RSComp p-value = 2.0e−04. Source data are provided as a Source Data file.

selective regions. These ROI analyses indicate that object context is not only represented in the PPA but also in other scene-selective and object-selective regions of high-level visual cortex. That said, it is notable that the effects for object2vec were numerically strongest in the PPA, supporting the idea that this region is particularly important for linking objects to their visual contexts.

We performed whole-brain analyses to examine the prediction accuracy of the object2vec and word2vec models across all voxels with reliable signal (see "Methods" section for details). The results of these analyses are shown in Fig. 6 (see Supplementary Fig. 4 for single-subject results). The findings were generally consistent with the results of the ROI analyses, with a cluster of relatively high prediction accuracy for the object2vec model overlapping with anterior PPA and extending into parahippocampal cortex. This cluster was strongest in the right hemisphere in a region of the ventral visual cortex where lateral PPA overlaps with a medial part of the object-selective pFs. There were also several other smaller patches of significant prediction accuracy for object2vec, including the scene-selective OPA and RSComp. For the word2vec model, a prominent cluster of significant prediction

accuracy was observed in a region of the ventral visual cortex, immediately adjacent to and overlapping with the largest cluster for the object2vec model in the right hemisphere. This cluster of significant prediction accuracy for word2vec appears to overlap with both pFs and PPA. Other clusters of high prediction accuracy for word2vec were observed on the lateral surface near object-selective LO (see also ROI results in Fig. 5). There were also several other patches of significant prediction accuracy for word2vec, including the scene-selective OPA and RSComp.

We next directly compared voxel-wise prediction accuracies for the object2vec and word2vec encoding models. We generated a preference map showing voxels with significantly higher prediction accuracy for either object2vec or word2vec (Fig. 7A). This analysis showed that object2vec generated higher prediction accuracies in a small cluster overlapping with the anterior portion of the PPA parcel in the right hemisphere, which is consistent with the findings from our ROI analyses. In contrast, word2vec generated higher prediction accuracies in more lateral regions of the ventral visual cortex overlapping with the object-selective pFs and LO.

**Category selectivity and co-occurrence models.** The results of the ROI and whole-brain analyses suggested a general trend for the image-based object2vec to be associated with scene-selective regions and the language-based word2vec to be associated with object-selective regions. To quantify this trend at the single-voxel level, we performed a follow-up analysis that compared the voxel-wise effects of the encoding models with measurements of scene-selectivity and object-selectivity from the functional localizer data. Specifically, we extracted voxel-wise responses to scenes and objects from a separate set of localizer data and computed activation difference scores for scenes minus objects. In the same voxels, we also computed accuracy difference scores for the encoding-model effects of object2vec minus the encoding model-effects of word2vec. We restricted our analyses to voxels that showed a significant effect in either the object2vec or word2vec encoding models, and we computed the correlation of the activation and accuracy difference scores across these voxels. This analysis showed a positive linear trend in three out of four subjects between the voxel-wise difference scores of scene and object selectivity and voxel-wise difference scores of object2vec and word2vec accuracy (Fig. 7B). This exploratory analysis suggests that voxels that are better explained by a model based on the co-occurrence statistics of objects in images tend to have greater scene selectivity, whereas voxels that are better explained by a model based on the co-occurrence statistics of object names in language tend to have greater object selectivity. However, this was a *post hoc*, exploratory analysis and the trend was not detected in all subjects. We have therefore not drawn any strong conclusions from these findings.

**Object spatial properties and co-occurrence statistics.** Previous work has shown that the response of the PPA to objects is modulated by their spatial properties, such as real-world size[34,35] and spatial stability[20,21]. These findings suggest that the PPA encodes not only the spatial layout of visual scenes[17], but also spatial information that is inherent in individual objects or evoked by their perception. We expected that the natural statistics of object co-occurrence captured by object2vec would covary with the spatial properties of objects. Indeed, the covariance of these co-occurrence statistics with higher-level object properties is precisely what makes these co-occurrence statistics useful. However, we expected that object2vec would also incorporate additional associative information that is not reducible to similarities in object spatial properties. To examine this issue, we collected behavioral ratings for real-world size and spatial stability, and we

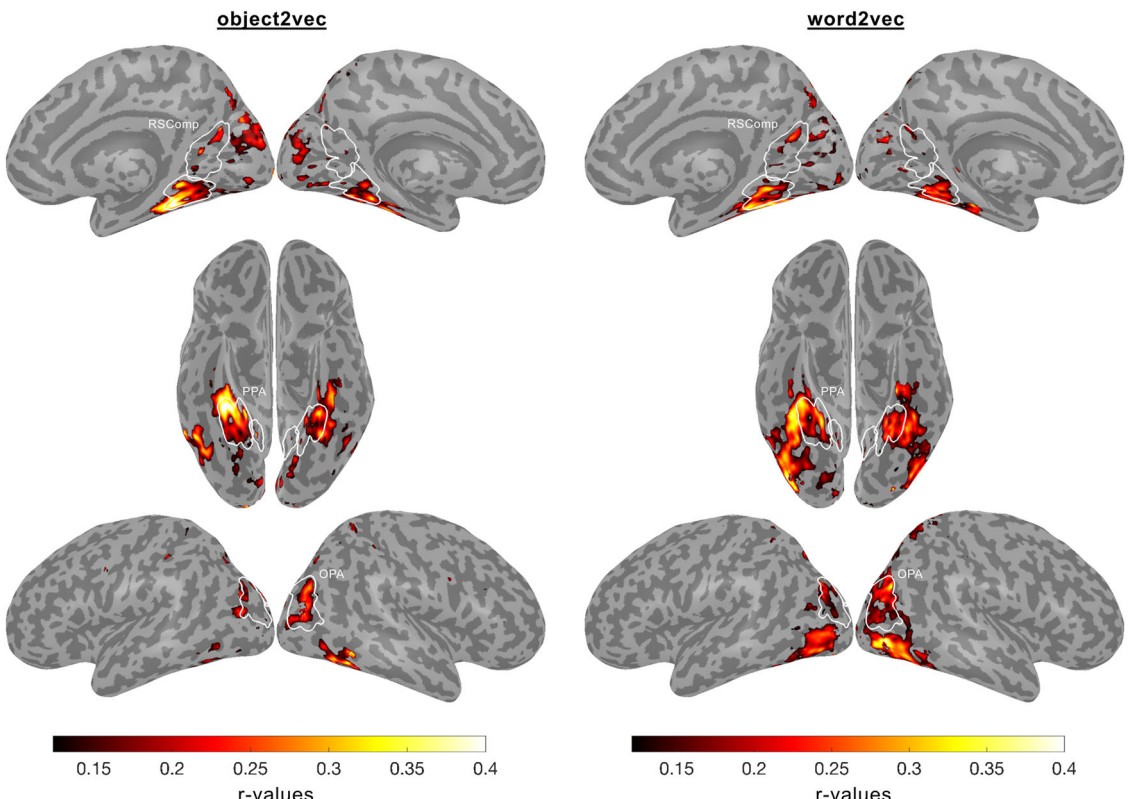

**Fig. 6 Encoding models of visual and linguistic context predict fMRI responses in distinct but overlapping regions of high-level visual cortex.** Voxel-wise encoding models were estimated and assessed using the 9-fold cross-validation procedure described in Fig. 3. These analyses were performed for all voxels with split-half reliability scores greater than or equal to $r = 0.1841$, which corresponds to a one-sided, uncorrected $p$-value of 0.05 (see split-half reliability mask in Supplementary Fig. 8). Encoding model accuracy scores are plotted for voxels that show significant effects ($p < 0.05$, FDR-corrected, one-sided permutation test). The left panel shows prediction accuracy for the image-based object2vec encoding model, and the right panel shows prediction accuracy for the language-based word2vec encoding model. There is a cluster of high prediction accuracy for the object2vec model overlapping with the anterior portion of the PPA, mostly in the right hemisphere. There are also several other smaller clusters of significant prediction accuracy throughout high-level visual cortex, including clusters in the OPA and RSComp. The word2vec model produced a cluster of high prediction accuracy that partially overlapped with the significant voxels for the object2vec model in the right PPA but extended into a more lateral portion of ventral temporal cortex as well as the lateral occipital cortex. ROI parcels are shown for scene-selective ROIs. PPA parahippocampal place area, OPA occipital place area, RSComp retrosplenial complex.

used these ratings as regressors in an encoding model of spatial properties.

We first performed ROI analyses to confirm that the spatial properties encoding model predicted responses in the PPA and other scene-selective ROIs (Fig. 8; see Supplementary Fig. 5 for single-subject ROI results and see Supplementary Fig. 6 for whole-brain results). As expected, the spatial-properties model generated significant prediction accuracies in all scene-selective ROIs, and it also generated weaker but significant prediction accuracies in EVC and the object-selective pFs.

We next directly compared voxel-wise prediction accuracies for object2vec and the spatial properties model. We generated a preference map showing voxels with significantly higher prediction accuracy for one model vs. the other (Fig. 9). This analysis showed that the spatial properties model generated higher prediction accuracies in portions of all three scene-selective ROIs, including most of the PPA. In contrast, object2vec generated higher prediction accuracies in a cluster that overlapped with the anterior PPA and extended into non-PPA portions of the parahippocampal cortex.

**Principal components of voxel tuning.** Finally, we performed an exploratory analysis to visualize how tuning for the object2vec and word2vec models varied across voxels. For each model

(object2vec and word2vec), we applied PCA to the regression weights for all voxels that had significant prediction accuracies. We visualized the first four principal components (PCs), by projecting each object category onto these PCs. Figure 10 shows the object categories color-coded according to their score on each PC. A common theme across most PCs is the broad distinction between indoor and outdoor environments. Additionally, each PC emphasizes different subsets of indoor and outdoor objects. For example, PC3 differentiates electronics and appliances from other indoor objects (e.g., microwave and computer vs. soap and napkin), and it differentiates between man-made and natural items found in outdoor scenes (e.g., sea and sand vs. sidewalk and car). However, these PCs also contain complex mixtures of object clusters that are not easily interpretable as representing any single semantic dimension. The semantic complexity of these PCs may reflect the mixed selectivity that arises from encoding a large set of object contexts in a small number of representational dimensions (i.e., the 8 dimensions of object2vec). A similar visualization is provided for the word2vec model in Supplementary Fig. 7. These exploratory visualization analyses can help us to develop intuitions about cortical tuning for object context, but as with many data-driven analyses, it is important to keep in mind that our interpretations of these visualizations are post hoc and have not been statistically tested.

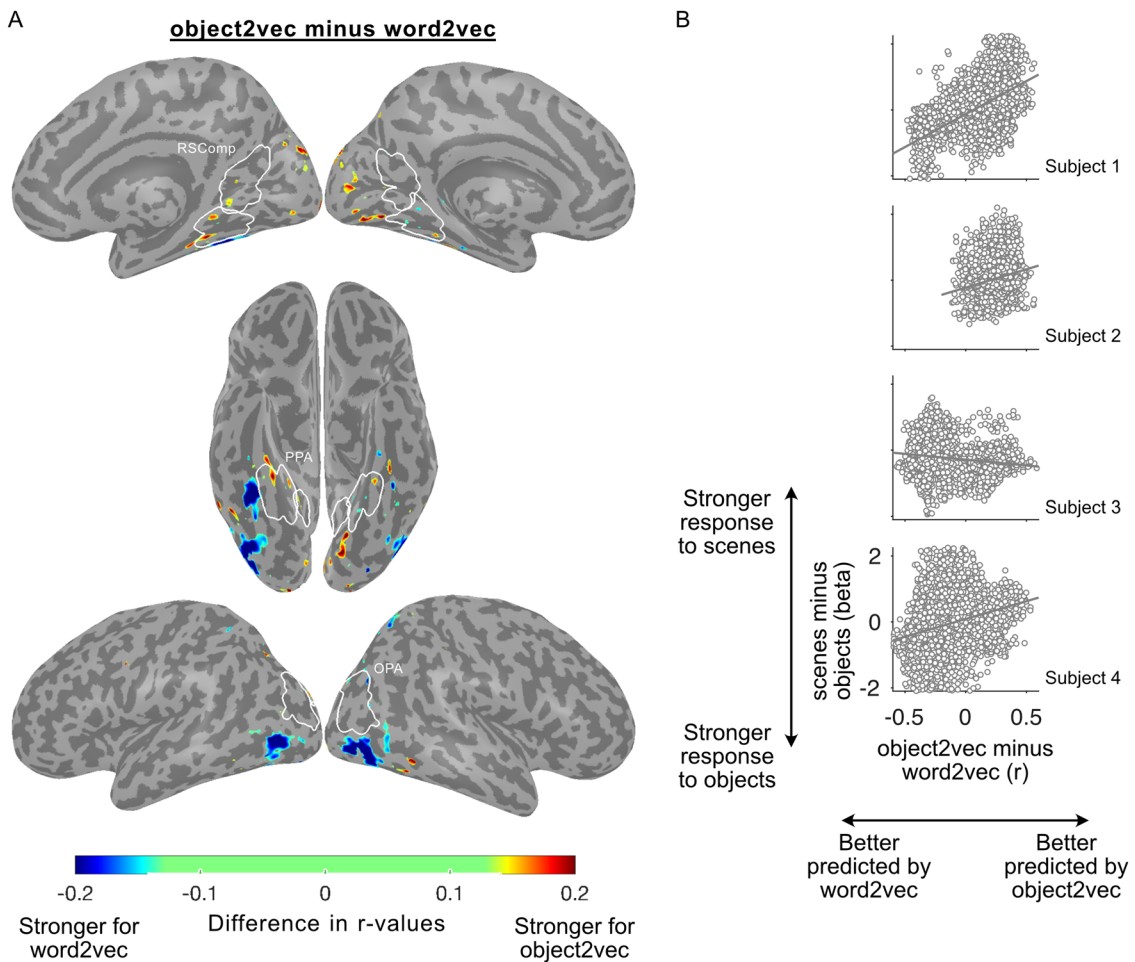

**Fig. 7 Whole-brain voxel-wise preference map for visual vs. linguistic context and relationship to category selectivity. A** This preference map shows differences in model performance for the object2vec (visual context) encoding model and the word2vec (linguistic context) encoding model. Voxel-wise difference scores were calculated by subtracting prediction accuracy for the word2vec model from prediction accuracy for the object2vec model. Warm colors indicate voxels that were better explained by object2vec and cool colors indicate voxels that were better explained by word2vec. Difference scores are plotted for voxels that show a significant preference for either model ($p < 0.05$, FDR-corrected, one-sided permutation test). There are clusters of significantly greater prediction accuracy for object2vec relative to word2vec overlapping with anterior PPA, mostly in the right hemisphere. There are clusters of significantly greater prediction accuracy for word2vec relative to object2vec in regions lateral to the PPA, extending to the lateral occipital cortex. ROI parcels are shown for scene-selective ROIs. PPA parahippocampal place area, OPA occipital place area, RSComp retrosplenial complex.
**B** Category selectivity of individual voxels is plotted as a function of their encoding-model preferences. For each subject, all voxels with significant effects for either the image-based object2vec encoding model or the language-based word2vec encoding model are plotted. The x axis plots the difference in prediction accuracy between the object2vec encoding model and the word2vec encoding model. The y axis plots the difference in activation between scenes and objects, based on data from a separate set of functional localizer runs. Positive linear trends suggest that voxels that are better predicted by object2vec than word2vec tend to be more scene-selective and voxels that are better predicted by word2vec than object2vec tend to be more object-selective (Pearson r values = 0.51, 0.20, −0.17, 0.29 for subjects 1–4). Source data are provided as a Source Data file.

## Discussion

The goal of this study was to determine if object representations in visual cortex reflect the statistical regularities of object co-occurrence in the visual environment. We first developed an objective model of the latent statistical structure of object co-occurrence using an unsupervised machine-learning algorithm applied to a large set of annotated images of real-world scenes. We then compared the representations of this model with fMRI activity evoked by single objects shown in isolation. We found that responses of voxels in scene-selective visual cortex—most strongly, the anterior PPA and adjoining portions of para-hippocampal cortex—were predicted by the co-occurrence sta-tistics of objects in visual scenes. In contrast, the representations of a language-based co-occurrence model for object names better predicted the responses of voxels in object-selective cortex. These findings indicate that scene-selective cortex links objects with

representations of their visual contexts and that anterior PPA is a particularly important locus for performing this operation. They further suggest that the regularities of vision and language map onto partially distinct components of cortical object processing.

Our results provide insights into how visual context is repre-sented in the brain, a longstanding issue in visual neuroscience. Previous behavioral studies have shown that the associations between objects and their typical contexts can influence perfor-mance on perceptual tasks, including object recognition, scene recognition, and visual search[1–4]. For example, when an object is shown in an unusual context (e.g., a pig in a cathedral), subjects are slower and less accurate at recognizing both the foreground object and the background scene[1]. One possible mechanism that could underlie these behavioral effects is a representation of an object's statistical associations to other objects and scene ele-ments. Previous neuroimaging work on contextual associations

## spatial properties

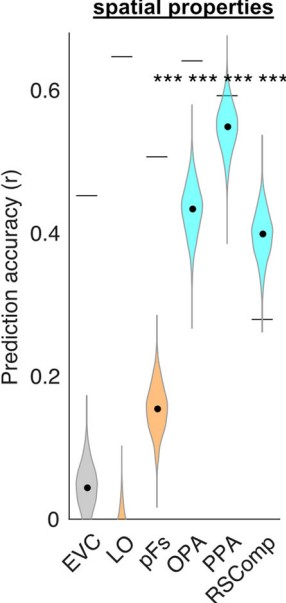

## object2vec minus spatial properties

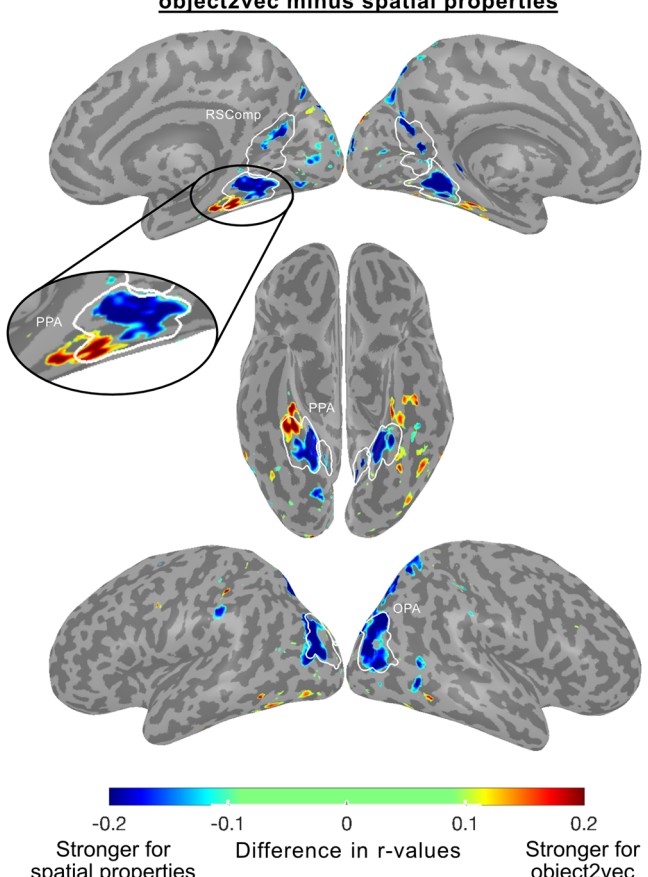

**Fig. 8 Encoding model of object spatial properties predicts fMRI responses in functionally defined ROIs.** This plot shows the average prediction accuracies for encoding models in voxels from multiple regions of interest using object-wise spatial property ratings as regressors (i.e., real-world size and spatial stability). Object-selective ROIs are plotted in orange and scene-selective ROIs are plotted in cyan. The violin plots show the mean prediction accuracies (central black dots) and bootstrap standard deviations. The gray lines above each violin plot indicate the average voxel-wise split-half reliability of the fMRI responses in each ROI. As expected from previous results, object-wise spatial properties were good predictors of fMRI responses in scene regions. EVC early visual cortex, LO lateral occipital, pFs posterior fusiform, OPA occipital place area, PPA parahippocampal place area, RSComp retrosplenial complex. ***$p < 0.001$, uncorrected, one-sided permutation test. Exact $p$-values: EVC $p$-value = 1.4e−01; LO $p$-value = 8.7e−01; pFs $p$-value = 2.0e−04; OPA $p$-value = 2.0e−04; PPA $p$-value = 2.0e−04; RSComp $p$-value = 2.0e−04. Source data are provided as a Source Data file.

has found evidence consistent with this hypothesis. Specifically, it has been shown that objects with strongly associated scene contexts (e.g., stoves are strongly associated with kitchen contexts), elicit an increased mean fMRI signal in the PPA and RSComp, two regions of scene-selective visual cortex[11,19,29]. However, the implication of these results has been unclear, because other ratings of object properties, such as real-world size and spatial stability, have been found to covary with contextual-association ratings and explain similar or more variance in fMRI responses[20,21]. Moreover, almost no previous studies have investigated the fundamental question of whether object-evoked responses in visual cortex represent the underlying multi-dimensional structure of object context (i.e., which objects are associated with each other). One previous study examined multivoxel patterns for scenes and objects and found no relationship between contextually related objects in the PPA[36]; however, this study only tested eight object categories and used simple pattern classification methods that may have been less sensitive to contextual effects.

The current experiment overcomes the limitations of previous studies by explicitly modeling the statistics of object co-occurrence in the visual environment and relating these statistics to object-evoked fMRI responses. We addressed the challenge of objectively quantifying the real-world statistics of object context by leveraging a large data set of densely labeled images that

**Fig. 9 Whole-brain voxel-wise preference map for visual context (object2vec) vs. object spatial properties.** This preference map shows differences in model performance for the object2vec encoding model and the spatial-properties encoding model. Voxel-wise difference scores were calculated by subtracting prediction accuracy for the spatial properties model from prediction accuracy for the object2vec model. Warm colors indicate voxels that were better explained by object2vec and cool colors indicate voxels that were better explained by spatial properties. Difference scores are plotted for voxels that show a significant preference for either model ($p < 0.05$, FDR-corrected, one-sided permutation test). There are clusters of significantly greater prediction accuracy for object2vec relative to spatial properties in the anterior PPA and in the parahippocampal cortex beyond the anterior PPA boundary. There are clusters of significantly greater prediction accuracy for spatial properties relative to object2vec in posterior PPA and the other scene-selective ROIs (OPA and RSComp). ROI parcels are shown for scene-selective ROIs. PPA parahippocampal place area, OPA occipital place area, RSComp retrosplenial complex.

was originally developed for training semantic segmentation models in computer vision[27]. We analyzed the annotations for these images using object2vec, which is a modified version of the word2vec algorithm from computational linguistics[23,24]. An important aspect of this modeling procedure is the fact that we explicitly analyzed the co-occurrence of object categories in scenes. This was possible because we applied a text-based machine-learning algorithm to image annotations. In comparison, a learning algorithm that took pixel values as inputs, such as a convolutional neural network, might also learn the natural statistics of object co-occurrence[37], but this co-occurrence information would be latent in the model's internal representations and would thus be difficult to observe and quantify.

A similar approach of combining methods from computational linguistics with image annotations was previously used to model

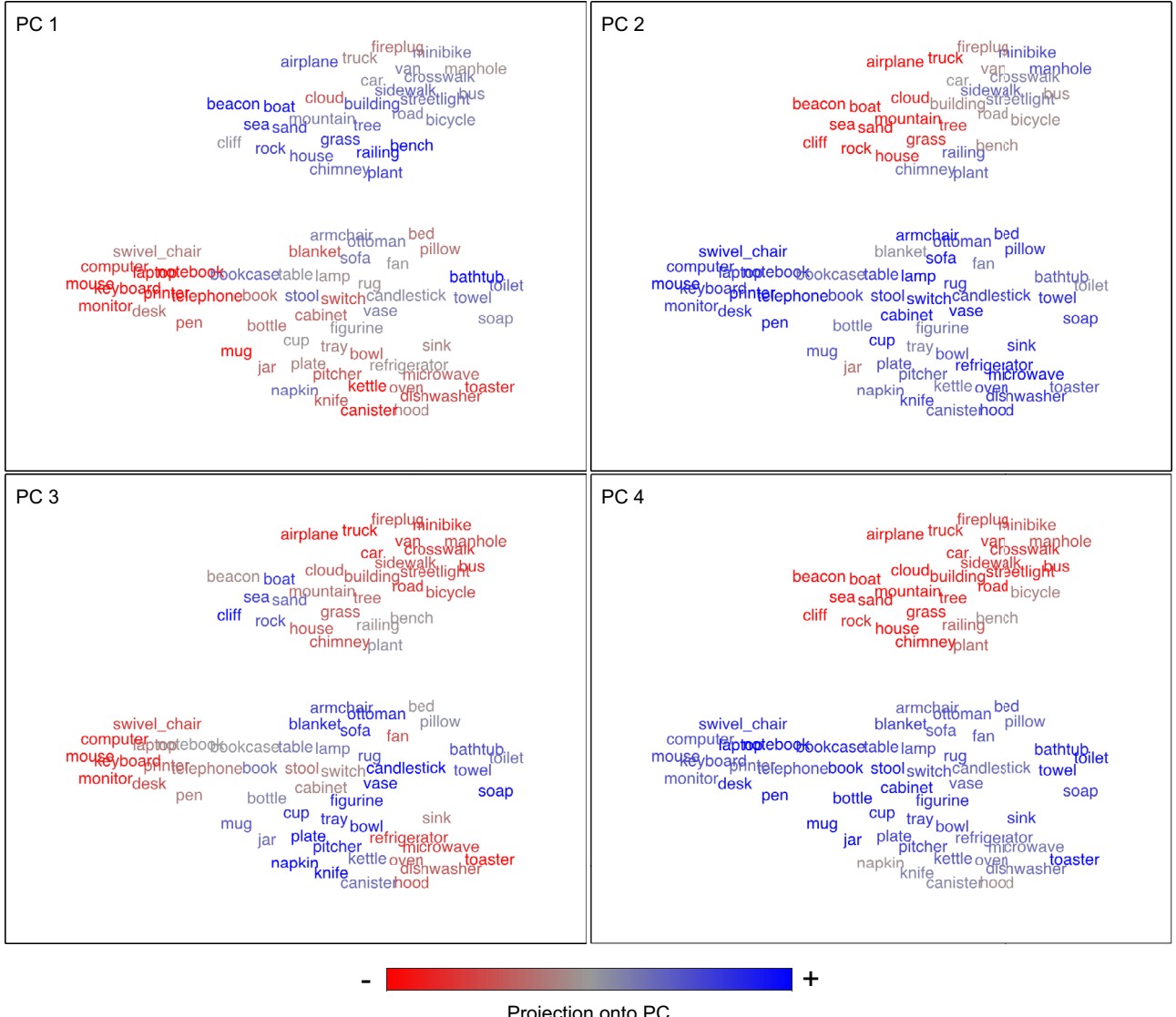

**Fig. 10 Principal components of voxel tuning for visual object context.** Principal components analysis was used to examine variance in encoding-model regression weights across voxels. This plot illustrates the first four principal components (PCs) of the regression weights for the image-based object2vec encoding model, using all voxels with significant prediction accuracies. The 81 object categories from the fMRI experiment were projected onto each PC (as indicated by the color coding from blue to red). Thus, the PC score for each object is conveyed by color rather than spatial position. The spatial arrangement of the object categories is the same as in the tSNE plot of Fig. 1. We used this spatial arrangement to facilitate comparison with the embeddings before they were projected onto the encoding-model PCs and to facilitate comparisons of different PCs. At a coarse level, most PCs broadly distinguish between items found in indoor and outdoor environments. At a more fined-grained level, each PC organizes the objects into distinct sets of clusters, some of which appear to be semantically interpretable. For example, PC2 differentiates electronics and appliances from other indoor objects (e.g., microwave and computer vs. soap and napkin), and it also differentiates between man-made and natural items found in outdoor scenes (e.g., sea and sand vs. sidewalk and car). However, these PCs also contain complex clusters of objects that do not appear to have intuitive semantic interpretations. This semantic complexity may reflect the mixed selectivity that arises from encoding a large set of object contexts in a small number of representational dimensions (i.e., the 8 dimensions of object2vec). Source data are provided as a Source Data file.

scene categorization behaviors[38] and to show that contextual information derived from object co-occurrence statistics is at least partially correlated with semantic information derived from the co-occurrence statistics of natural language[26]. A similar method was also used in an earlier fMRI study to model the influence of object co-occurrence statistics on cortical responses to scenes[25]. Our work is related but addresses a different set of questions. We set out to understand how visual cortex links objects with representations of their contexts. For this purpose, it was crucial that we used isolated single objects as stimuli, rather than full scenes (for which contextual information would already be

present in the visual display). This approach allowed us to identify a set of cortical regions that encode the contextual information associated with visual objects, even in the absence of a surrounding scene.

It is notable that the strongest evidence for contextual coding was found in the anterior PPA, extending into adjacent parts of the parahippocampal cortex. The object2vec model predicted the greatest amount of variance in this region, and this was also the only part of the PPA where object2vec explained more variance than a model based on object spatial properties (i.e., real-world size and spatial stability). Previous fMRI studies of visual

context typically found results throughout the PPA and other scene-responsive regions[11,15], although one study reported a more anterior parahippocampal locus (outside of the PPA) for non-spatial associations between objects[16]. Our findings are consistent with other data implicating anterior PPA in the coding of abstract information related to scenes and contexts. For example, a previous study found that multivoxel activation codes in this region generalized across interior and exterior views of the same building, indicating some degree of abstraction related to building identity[39]. Functional connectivity data also suggest the possibility of a division between anterior and posterior PPA, with posterior PPA more connected to areas implicated in scene perception, such as the OPA and early visual cortex, and anterior PPA more connected to regions involved in spatial perception and memory, such as RSComp, caudal inferior parietal lobule, and medial temporal lobe[30,40]. Beyond the PPA, we found weaker though significant object context effects in scene-selective RSComp and OPA, and in object-selective pFs. Thus, although anterior PPA may be the most crucial locus, contextual processing appears to involve a broad network of high-level visual regions.

In addition to looking at representations of object co-occurrence statistics in visual scenes, we also examined the co-occurrence statistics of object names in linguistic corpora. We found an interesting and unexpected relationship between the modality of the distributional models (i.e., vision or language) and the category selectivity of the regions that were best predicted by these models. Specifically, we found that language-based word2vec produced stronger prediction accuracies in object-selective pFs and LO, whereas image-based object2vec produced stronger prediction accuracies in the anterior portion of scene-selective PPA. Previous work has shown that language-based models of distributional statistics can predict object-evoked responses in both fMRI and magnetoencephalography[41,42], but no previous studies have examined the possibility that responses in the object-selective cortex might be better predicted by the distributional statistics of language than the distributional statistics of vision.

What might account for this pattern of findings? One relevant consideration is that language-based word2vec likely contains more taxonomic information (e.g., What class of object is this?), whereas image-based object2vec emphasizes thematic information (e.g., What places and events is this object found in?). Indeed, previous findings from computational linguistics suggest that standard models of distributional semantics often perform better on tests of taxonomic knowledge than tests of thematic knowledge[43]. Thus, one possibility is that the regions predicted by word2vec and object2vec encode different domains of abstract semantic associations that are not perceptual in nature. In previous work that is consistent with this more semantically oriented interpretation, posterior fusiform gyrus and parahippocampal cortex have been shown to be involved in the processing of object semantics, even when the experimental stimuli are words rather than images (but note that these studies did not specifically compare taxonomic and thematic associations)[44,45]. However, a different, but not mutually exclusive, possibility follows from the fact that objects from similar taxonomic categories tend to have similar shapes. Insofar as the word2vec embeddings for object terms in language capture information about object shape, they may be better predictors of perceptual responses in object-selective visual cortex. Our findings cannot adjudicate between these possibilities. Thus, the role of abstract semantic associations and perceptual features in explaining these contextual representations should be explored in future work.

In addition to advancing our understanding of how visual and linguistic context are represented in the brain, our results also have implications for understanding cortical function more generally. A fundamental theory in visual neuroscience is that neurons in the visual system are tuned to the natural statistics of the sensory environment[46,47]. Specifically, it is thought that neural response preferences are shaped through evolutionary and developmental processes to optimize "efficiency"—that is, to convey as much useful information as possible with the limited computational resources of the brain. One potential way to improve the efficiency of a neural population code is to match neural tuning properties to the natural statistics of sensory stimuli, such that the proportion of neurons tuned to a particular stimulus is related to the frequency of that stimulus in sensory experience. These ideas have been investigated in the context of low-level vision by calculating sensory statistics from readily available image properties, such as pixel intensities[48–50]. The natural statistics of higher-level sensory information are also likely to be relevant for understanding the functions of visual cortex, but a major challenge in testing this idea is the need for large-scale annotated stimulus sets and quantitative approaches for modeling their statistical structure. By combining annotated image datasets with tools from computational linguistics, as we do here, vision researchers can benefit from decades of work that has sought to model the latent structure of language from the distributional statistics of words in text[28].

The approach of examining the high-level statistical regularities of objects and scenes may provide insight into longstanding questions about cortical function. In the case of scene-selective cortex, previous work has shown that scene regions respond to several high-level properties of objects, including contextual-association strength, real-world size, landmark suitability, spatial definition, and interaction envelope[11,21,34,35,51]. Scene-selective regions also respond to several mid-level properties of images, including high spatial frequencies, rectilinearity, and cardinal orientations[52–55]. Many of the high-level object properties appear to covary in the visual environment and explain similar variance in fMRI responses[20], and they may also covary with mid-level image properties[56,57]. One question that is often raised is whether any of these high-level or mid-level stimulus properties is of primary importance in explaining the functions of the scene-selective cortex. Although it is possible that cognitive demands require scene-selective regions to be specialized for processing a specific and limited class of scene features, an alternative possibility is that these regions are tuned to the covariance of features in real-world scenes, and should thus show responses to a variety of stimulus properties, ranging from high-level to low-level. In this view, the representations of scene-selective cortex may reflect a statistically efficient basis set from which a large number of behaviorally relevant aspects of scenes can be decoded. Our current data support this idea by showing that a large portion of the explainable variance in PPA can be accounted for by a low-dimensional model based on the natural statistics of objects in scenes. Furthermore, this idea is supported by the overlapping explained variance that we observed for the object2vec and object spatial property models, which suggests that the natural statistics of object co-occurrence are related to physical characteristics of the objects, such as their real-world size. We expect that a number of other high- and mid-level object and scene properties would be linked to these statistics, consistent with the hypothesis that cortical tuning to the statistical regularities of scenes produces representations that are informative to a broad range of stimulus properties. Similar statistical principles may underlie the cortical representations that support other high-level cognitive functions, such as object categorization[58], face recognition[59], and reinforcement learning[60].

This perspective suggests several questions for future investigations. One open question is whether the visual cortex contains

representations that reflect the natural statistics of object locations in images. Here we examined object co-occurrence independent of object location, but it is known that the visual system also takes advantage of associations between objects and spatial locations[61] and that perceptual behaviors and cortical representations are sensitive to violations of the typical positions of objects[2,62–64]. Future investigations could utilize annotated image databases to explicitly quantify these spatial statistics and to examine how they relate to the object representations of visual cortex. A second open question relates to how the statistics of object co-occurrence are learned and incorporated into the representations of visual cortex. One possibility is that these statistics are initially encoded in memory regions of the medial temporal lobe and then transferred to visual cortex[65]. A third open question is whether object2vec or similar image-based approaches could be merged with word embeddings from computational linguistics to create richer representational models of object semantics[66]. Our findings suggest that object2vec could complement language-based representations by bringing in perceptual information that is not easily learned from language corpora. More broadly, we believe that future research on high-level vision would benefit from a greater emphasis on the statistical basis of cortical representations and their utility for behaviorally relevant computational goals.

## Methods

**Subjects**. Four healthy subjects (two female, ages 30, 32, 32, and 34) from the University of Pennsylvania community participated in the experiment. All had normal or corrected-to-normal vision and provided written informed consent in compliance with procedures approved by Institutional Review Boards at the University of Pennsylvania and Johns Hopkins University.

**MRI acquisition**. Participants were scanned on a Siemens 3.0T Prisma scanner using a 64-channel head coil. We acquired T1-weighted structural images using an MPRAGE protocol (TR = 2200 ms, TE = 4.67 ms, flip angle = 8°, matrix size = 192 × 256 × 160, voxel size = 0.9 × 0.9 ×1 mm). We acquired T2*-weighted functional images sensitive to blood oxygenation level-dependent contrasts using a multiband acquisition sequence (TR = 2000 ms for main experimental scans and 3000 ms for localizer scans, TE = 25 ms, flip angle = 70°, multiband factor = 3, matrix size = 96 × 96 × 81, voxel size = 2 × 2 × 2 mm).

**Stimuli**. Subjects viewed images of individual objects presented on textured backgrounds. The objects were from 81 different categories, with 10 unique images per category, resulting in a total of 810 experimental images. The stimuli were guided by the labeled segmentation classes in ADE20K, which included not only independent objects but also fixed scene elements, like mountains and roads. We use the term "objects" here to refer to scene elements that one would identify if asked to name the things in a picture (as in the labeling procedure for ADE20K). A benefit of this approach is that it is unbiased in the selection of nameable scene elements. An alternative approach of restricting the study to independent objects may run the risk of not generalizing to the much richer set of scene elements that are important to contextual processing in natural visual experience. It is also worth noting that previous related studies have similarly treated fixed scene elements as objects. For example, the Bar & Aminoff[11] study on contextual associations included objects such as barn, roulette table, and windmill, and the Stansbury et al.[25] study on object co-occurrence included objects such as sky, sand, and floor.

Images of objects were obtained by searching for the object category terms on Google Images and downloading high-quality images in which the object was prominent. We then manually isolated the object from the image background using Adobe Photoshop. All images were cropped around the isolated object and then resized to 600 × 600 pixels with the original aspect ratio preserved (thus, the shorter axis of the image was padded with transparent pixels).

We placed the objects over complex, textured backgrounds (Fig. 2). These background textures were generated using a procedure that was designed to systematically reduce the similarity between our models of interest (i.e., object2vec and word2vec) and a model of low-to-mid-level perceptual features (i.e., AlexNet trained on ImageNet). We used representational similarity analysis (RSA) to assess the similarity of these models. First, representations were generated for all stimuli as the vectorized outputs of each model (i.e., object2vec embeddings, word2vec embeddings, and unit activations of AlexNet). For AlexNet, representations for the 10 images within each object category were averaged to create a single representation per category. Next, representational dissimilarity matrices (RDMs) were created by calculating all pairwise comparisons of the representational vectors for the 81 object categories. Representational dissimilarities between categories

were measured using Pearson distance (one minus the Pearson correlation coefficient). We then assessed the similarity between models by calculating the Spearman correlation of their RDMs, which we refer to as an RSA correlation.

Background textures were created from composites of DeepDream visualizations for all units in convolutional layer 5 of AlexNet using the MATLAB function deepDreamImage. DeepDream visualizations were initiated with random pixel values and optimized through gradient ascent using a multi-resolution image pyramid and Laplacian Pyramid Gradient Normalization (3 pyramid levels, scaling of 1.4 between pyramid levels, 10 iterations per level). We created composite textures by averaging the visualizations from three channels sampled randomly with replacement and then adjusting the overall luminance to 90 in Lab color space. We used a stochastic optimization procedure to synthesize composite background textures for each object image that minimized the RSA correlations of the CNN layers with both the object2vec and word2vec models. Our optimization criterion, which we sought to minimize, was the maximum correlation coefficient ($r_{max}$) across all RSA comparisons between the models of interest (i.e., object2vec and word2vec) and the convolutional layers of AlexNet (layers 1–5). We began by randomly synthesizing 3,000 composite background textures and generating 1000 random assignments of these backgrounds to the 810 object images. From these, we selected the set of object-background pairings with the lowest $r_{max}$. We then implemented an iterative optimization procedure in which the image backgrounds were individually altered to gradually reduce $r_{max}$. On each iteration, a new randomly synthesized background texture was generated for a single randomly selected object, and $r_{max}$ was re-calculated. If $r_{max}$ was lower than the current best value, the new object-background pairing from that iteration was retained; otherwise, it was discarded. This procedure was stopped shortly after $r_{max}$ dropped below 0.05.

We also created a set of warped object stimuli by applying a diffeomorphic transformation to the 810 isolated object images[67]. The diffeomorphic warping was implemented over 5 iterations with a maximum distortion of 30. These warped objects served as target stimuli for a category-detection task in the fMRI scanner (see Supplementary Fig. 1 for examples).

**Procedure**. We collected fMRI data while subjects viewed images of objects on textured backgrounds and performed a category-detection task (implemented in PsychoPy 1.84). Subjects were asked to fixate on a central cross that remained on the screen throughout the scan and press a button whenever the stimulus was a warped object. Successful performance on this task required subjects to attend to each stimulus but did not require them to think about the mnemonic associations of the objects. Stimuli were presented in a mini-block design. In each block, 5 images from the same object category were presented in rapid succession. Each image was shown for 500 ms, and consecutive images were separated by a 500 ms inter-stimulus interval, during which a random background texture was shown, which differed from the backgrounds that the objects were shown on. After the fifth image in each block, a gray screen with a central cross appeared for 1.5 s before the start of the next block. Each block was thus 4.5 s in length, with a minimum inter-block interval of 1.5 s. Every run also included 6 randomly placed null events of 4 s each, which always occurred between blocks.

Consistent with previous studies using encoding model analysis methods, we sought to obtain a large amount of fMRI data from each participant, rather than data from a large number of participants. The experiment was designed to be completed in 4 sessions, each taking place on different days; however, two subjects chose to split up the scanning sessions over more than 4 days. For each planned session, we randomly divided the 81 object categories into 9 folds of 9 categories each. These folds were presented in separate runs, randomly ordered. Runs contained 6 mini-blocks for each of the 9 categories; images were assigned to mini-blocks by creating 3 different 2-fold splits for the 10 images in each category. Each run also contained 1–3 warped-object mini-blocks, in which a single warped object image was shown among 4 normal images from the same object category. The categories for the warped-object mini-blocks were randomly chosen from the 9 categories in the run. The order of the mini-blocks within the run was randomized, as was the order of the images within each mini-block, subject to the restriction that mini-blocks from the same category were never shown more than twice in a row. Runs were ~6.3 min in length, with the exact length depending on the variable number of warped-object mini-blocks (i.e., 1–3).

In two of the scanning sessions, additional fMRI data were collected for defining functional ROIs (described in the section Functional localizer and regions of interest). In another two scanning sessions, we also collected 11 min of fMRI data while subjects viewed a continuous natural movie (from the show Planet Earth). These movie data were collected for use in hyperalignment[68], and they were not analyzed for the current project.

**fMRI preprocessing**. fMRI data were processed and modeled using SPM12 (Wellcome Trust Centre for Neuroimaging, London, UK) and MATLAB (R2019b Mathworks). Functional images for each participant were realigned to the first image, co-registered to the structural image, warped to Montreal Neurological Institute (MNI) standard space, and spatially smoothed with a 6-mm full-width-half-maximum isotropic Gaussian kernel. Voxel-wise responses to each object category were estimated using a general linear model, which included regressors for each of the object categories and the target (warped-object) mini-blocks in each

run, and a regressor for each run. Each regressor consisted of a boxcar function convolved with a standard hemodynamic response function. Low-frequency drifts were removed using a high-pass filter with a cutoff period of 128 s, and temporal autocorrelations were modeled with a first-order autoregressive model. Voxel-wise beta values were z-scored across the 9 object categories within each run. These beta values were used as object category responses for all additional analyses.

**Voxel-wise reliability estimates**. We calculated noise-ceiling estimates based on the split-half reliability of voxel-wise responses to the 81 object categories. For each voxel, split-half reliability was calculated as the Pearson correlation between the mean responses to the 81 object categories in odd and even runs. The resulting reliability estimates were used to select voxels for further analyses[69]. Only voxels whose average split-half reliability across subjects was greater than or equal to $r = 0.1841$, corresponding to an uncorrected $p$-value of 0.05, were included in ROIs (see next section), or plotted in whole-brain maps (Supplementary Fig. 8).

**Functional localizer and regions of interest**. BOLD responses in two additional scan runs were used to define functional ROIs. The localizer protocol included 4.5-s blocks of scenes, faces, objects, and scrambled objects. Images were presented for 225 ms with a 75-ms inter-stimulus interval, and subjects performed a one-back repetition-detection task on the images. We identified the three scene-selective ROIs (PPA, OPA, and RSComp) based on a contrast of scenes > objects implemented in a general linear model with a group-based anatomical constraint of category-selective activation derived from a large number of localizer subjects in our laboratory ($n = 42$)[39,70]. Using a similar parcel-based approach, we defined two object-selective ROIs (LO and pFs) based on a contrast of objects > scrambled objects, and we defined EVC with the contrast of scrambled objects > scenes. Bilateral ROIs were defined by selecting the top 50 voxels from the parcel in each hemisphere based on the localizer contrast (i.e., the difference in beta values across conditions), excluding voxels that did not meet the split-half reliability criterion described above. For the anterior and posterior PPA ROIs, we restricted the full PPA parcel to either its anterior or posterior third and selected 25 voxels per hemisphere from each third.

**Representational models**. Our analyses made use of representational models based on statistical relationships between objects in natural images (object2vec) and linguistic corpora (word2vec), and a representational model based on the spatial properties of objects. The three representational models are described below.

**Object2vec**. Using tools from computational linguistics, we developed an approach for characterizing the latent statistical structure of object co-occurrence in natural images, which we refer to as object2vec. To train this model, we used annotated images from the ADE20K data set: http://groups.csail.mit.edu/vision/datasets/ADE20K/. This data set contains 22,210 scene images that have been densely annotated by human observers such that every object within every scene is segmented and labeled. Across the entire scene image set, there are 3148 unique object segmentation labels[27]. ADE20K is ideal for our purposes because it provides a large and diverse set of image annotations that we can use to model the co-occurrence statistics of objects in natural images.

Our modeling approach was inspired by the field of computational linguistics, which has a long history of modeling the lexical-semantic properties of words based on their co-occurrence statistics in natural linguistic contexts[28]. These co-occurrence-based models are referred to as distributional-semantic models because they reflect the distribution of words in language (often written language). The general goal of these models is to learn word representations (i.e., vectors) that capture the contextual similarity of words—that is, representations in which words that occur in similar contexts are similar. A leading algorithm for learning these distributional-semantic representations from text corpora is word2vec[23,24]. Object2vec is based on the continuous bag of words (CBOW) version of word2vec, which seeks to identify word representations that can be used to predict a missing target word from its surrounding context words (e.g., the other words around it in a sentence).

We adapted the word2vec CBOW algorithm, as implemented in fastText[71], and applied it to the image annotations from ADE20K. Our implementation modifies how the context window is defined for word2vec. Typically, each word's context is defined as a weighted window spanning forward and backward from the position of the word in a sentence. This approach to defining the context window is not applicable to our image annotations, which can either be analyzed as a bag of words, containing object labels in an arbitrary order, or as a 2D segmentation map. Here we implemented the simpler bag-of-words approach by defining the context window for each object so that it includes all other objects in the same image (with equal weights for all context objects). Note that fastText allows for the possibility of learning representations for character n-grams (i.e., groups of letters within words), but this option was not applicable to our image annotations and was thus not used.

We first created an annotation data set to use as our training corpus. We converted each image into a list of object labels, without including repeats for objects that occurred multiple times in the same image. The object labels were cleaned up and simplified in several ways. In ADE20K, each object can be associated with a list of multiple possible names (e.g., "bathtub, bathing tub, bath, tub"), which we converted into a single name using the first term in the list (e.g., "bathtub"). For the 81 object categories in our experimental stimuli, we manually inspected the object labels in ADE20K to identify other possible labels that could be applied to these objects (e.g., "bowl" and "bowls", "notebook" and "notepad"). These label sets are included in Supplementary Table 1. For categories with multiple associated labels, we converted all instances to one identical label (e.g., all instances of "notepad" were converted to "notebook"). After this procedure, we were left with 2959 unique object labels across the entire data set.

We created the object2vec embeddings by running our modified version of the fastText CBOW algorithm on the ADE20K annotation corpus. These embeddings are learned through stochastic gradient descent and the dimensionality of the embeddings is set by hand before model training. We first created embeddings of different sizes, spanning from 10 to 90 dimensions in steps of 20. For each dimensionality, we created 100 versions using different random initializations. The models were trained for 1000 epochs with a negative sampling parameter of 20. We sought to identify the simplest possible model that captured the contextual representations of the objects in ADE20K. By converting the resulting object2vec representations into RDMs, we could perform RSA to compare the information contained in the embeddings at different dimensionalities. We averaged the RDMs for all 100 versions of each dimensionality and then performed pairwise correlations of the RDMS for all dimensionalities. We found that the lowest-dimensional model (10 dimensions) was highly correlated with each of the higher-dimensional models (mean $r$-value = 0.92, sd = 0.02). Because of the strong representational similarity of these models, we used the simpler 10-dimensional embeddings for all further analyses. We then performed PCA on the 10-dimensional embeddings to determine if they could be further reduced in dimensionality. We found that on average across all 100 versions, 8 dimensions were sufficient to explain over 90% of the variance. Finally, we concatenated all versions of the 10-dimensional embeddings and used PCA to extract latent dimensions that were commonly identified across different random initializations. We retained 8 principal components (PCs), which served as our object2vec representations.

**Word2vec**. In addition to our object2vec embeddings, which were trained on image annotations, we also examined word2vec embeddings trained on written language. We used a set of 300-dimensional word2vec embeddings that had been previously trained on the Google News data set (~100 billion words): https://code.google.com/archive/p/word2vec/. We downloaded the word2vec embeddings here: https://github.com/chrisjmccormick/word2vec_matlab. These embeddings were filtered to include only words that are present in WordNet, which removes many entries that are not useful for our analysis of object concepts, including misspellings and multi-word phrases. We then used PCA to reduce the dimensionality of these representations. The goal of reducing the number of word2vec dimensions was to avoid overfitting by our encoding models. We inspected the scree plot of explained variance for each PC and found an elbow at ~30 PCs, where explained variance began to level off (Supplementary Fig. 9). We, therefore, retained 30 PCs of the word2vec embeddings for all further analyses. It is worth noting that it is not straightforward to compare the cumulative explained variance of the object2vec and word2vec PCs. The word2vec PCs were derived from high-dimensional embeddings (i.e., 300 dimensions) and a data set of 207,147 words, whereas the object2vec PCs were derived from low-dimensional embeddings (i.e., 10 dimensions) and a data set of 2,959 objects. We later ensured that our findings were not contingent on the specific number of PCs retained. We observed similar results in the range of 20–50 PCs. Beyond 50 PCs, cross-validated accuracy of the encoding models decreased, due to overfitting. For each of the 81 object categories in our experiment, we created a list of its associated names. This list included all names in the full ADE20K label for the object (e.g., "bathtub, bathing tub, bath, tub") as well as plural forms and any additional names that were detected through manual inspection of the WordNet-filtered word2vec vocabulary (207,147 words). For each object category, we averaged the embeddings for all of its associated names. The resulting 30-dimensional vectors served as our word2vec representations.

*Object spatial properties*: We collected behavioral ratings for two spatial properties of the objects in our experiment: real-world size and spatial stability. These data were collected on Amazon Mechanical Turk. Separate experiments were run for each spatial property rating. For real-world size, subjects were asked to indicate the size of the object using reference objects on a picture scale. The reference objects for the answer choices were a key, a bagel, a shoe, a backpack, a chair, a bed, a car, and a building, which were assigned to values 1–8. For spatial stability, subjects were asked, How often do you expect the position of this object to change in everyday life? The answer choices were very often, often, occasionally, rarely, or never, which were assigned to values 1–5. We filtered the data to retain only high-quality subjects, because we could not be certain that online raters attended to the stimuli. To do this, we computed all pairwise correlations of subjects' ratings and calculated the mean pairwise correlation for each subject. We then removed any subject whose mean pairwise correlation was more than 1.5 standard deviations away from the mean pairwise correlations of all subjects. We first collected 52 subjects for the real-world size ratings and retained 45 subjects after data filtering. After observing that few subjects were removed by our data-filtering procedure, we reduced the number of subjects collected to 30 for the spatial-stability ratings and retained 28 subjects after filtering. The across-subject

split-half reliability was high for the retained subjects ($r = 0.99$ for real-world size and $r = 0.94$ for spatial stability). Each subject rated one image from each of the 81 object categories in random order. We calculated the mean rating for each property across all retained subjects to obtain a final set of two property ratings for each object category. We used these property ratings as separate regressors in voxel-wise encoding models for spatial properties.

**Voxel-wise encoding models**. We used voxel-wise encoding models to determine if the fMRI responses to our 81 object categories could be predicted from the representations of object2vec, word2vec, and object spatial properties. For each subject, voxel-wise responses to the 81 object categories were determined by averaging responses across all runs. Ordinary least squares regression was used to estimate weights that map representational features (e.g., object2vec) to these fMRI responses. We used 9-fold cross-validation to quantify the out-of-sample prediction accuracy for each voxel-wise model. This cross-validation design was built into our fMRI protocol. As described above (see "Procedure" section), the stimuli for each subject were randomly split into 9 folds of 9 object categories each, and the object categories from different folds were always shown in different runs. For each iteration of the cross-validation procedure, voxel-wise regression weights for the object2vec and word2vec dimensions were estimated using object categories from 8 of the 9 folds, and the estimated regression weights were applied to the object categories in the held-out fold to generate predicted fMRI responses.

We included nuisance regressors in our encoding models to account for low-level stimulus properties that could influence the estimated weights for our regressors of interest (i.e., object2vec and word2vec). Nuisance regressors were generated using the five convolutional layers of a convolutional neural network (AlexNet) trained on ImageNet[72]: http://www.vlfeat.org/matconvnet/models/imagenet-caffe-ref.mat. The goal of this procedure was to create a small set of nuisance regressors that explain prominent low-level image features while not drastically increasing the overall number of regressors and thus causing problems with overfitting. We ran the experimental stimuli through AlexNet to obtain activations from the final output of each convolutional layer and then concatenated these activations into a single vector for each stimulus. We averaged these AlexNet activation vectors across all 10 images for each object category and used PCA to reduce their dimensionality. Based on inspection of the scree plot, we retained 20 PCs, which is at the point where explained variance leveled off. In the cross-validation procedure, the nuisance regressors were only included in the training folds for the purpose of estimating regression weights—they were not included when applying the regression weights to generate predictions for the object categories in the held-out test fold. Thus, all encoding-model predictions were computed using only the representations of object2vec, word2vec, or the object spatial properties model. We validated that the 20 AlexNet PCs captured variance in the responses of the visual cortex and were thus reasonable to use as nuisance regressors that reflect low-level stimulus properties. To do this, we fit voxel-wise encoding models using only the AlexNet PCs as regressors. As expected, we found high prediction accuracies in large portions of visual cortex in each subject (Supplementary Fig. 10).

We used permutation tests to assess the statistical significance of prediction accuracy scores. To estimate the distribution of effects within each voxel under the null hypothesis, we randomly permuted the object category labels within each fold of the cross-validation design and then computed the Pearson correlation between the actual fMRI responses and the permuted version of the predicted responses. We repeated this procedure 5000 times. We compared the actual $r$-value in each voxel to the null distribution to generate $p$-values and then adjusted for false discovery rate (FDR)[73]. For group-level analyses, we averaged the actual prediction accuracy scores in each voxel across subjects and did the same for the $r$-values from all 5000 iterations of the permutation procedure to produce a null distribution for each voxel. For ROI analyses, we calculated the average $r$-value across all voxels in all subjects (or within single subjects) and did the same for the $r$-values from all 5000 iterations of the permutation procedure. We compared the actual $r$-value for each ROI to the permutation-based null distribution to generate $p$-values. For the ROI analyses, we also obtained bootstrap standard errors by randomly resampling the actual and predicted fMRI responses within each fold and then re-computing the voxel-wise correlations and ROI means. This bootstrap resampling procedure was performed 5000 times.

We also performed a permutation-based interaction test on the encoding-model accuracy scores for object2vec and word2vec in anterior and posterior PPA. For each subject, we subtracted the voxel-wise accuracy for word2vec from the accuracy for object2vec. We calculated the average of these differences scores across all voxels in all subjects for each ROI, and we subtracted the average in posterior PPA from the average in anterior PPA. The resulting statistic reflects the degree to which object2vec produced higher prediction accuracy than word2vec in anterior PPA relative to posterior PPA. We calculated the same interaction statistic for 5000 iterations of our permutation procedure, and we compared the actual interaction statistic to the permutation-based null distribution to generate a $p$-value.

**Preference maps**. We compared voxel-wise prediction accuracies for pairs of encoding models by subtracting the prediction accuracy of one model from the prediction accuracy of the other model in each voxel. The result is a preference map showing which voxels are better predicted by one model or the other[74]. Before computing difference scores of prediction accuracies, we first set any negative prediction accuracies to zero to avoid the possibility that apparent preferences for one model could be driven by negative correlations for the comparison model. We used a permutation procedure to assign $p$-values to each preference score. To estimate the distribution of difference scores within each voxel under the null hypothesis, we randomly permuted the object category labels within each fold of the cross-validation design, using the same permutation indices for both encoding models being compared. We computed the Pearson correlation between the actual fMRI responses and the permuted version of the predicted responses for each encoding model and then calculated the difference score of prediction accuracies for the two models. We repeated this procedure 5000 times. We compared the actual difference score in each voxel to the null distribution to generate $p$-values and then adjusted for false discovery rate (FDR). For the group-level result, we averaged the actual difference scores in each voxel across subjects and did the same for the difference scores from all 5000 iterations of the permutation procedure to produce a null distribution for each voxel.

**Principal component analysis of encoding-model regression weights**. We performed a post hoc exploratory analysis to examine variance in the encoding model regression weights across voxels. Each voxel has a set of corresponding regression weights for each encoding model (e.g., 8 regression weights per voxel for object2vec). We calculated the average regression weights across subjects for each voxel, and we then performed PCA on these average regression weights across all voxels that had significant effects in the whole-brain analysis. We performed this procedure separately for the object2vec and word2vec encoding models. We applied the loadings for the first four principal components to the original regressors and generated PC scores for each object category in our experiment. To visualize these PCs, we color-coded the $t$-distributed stochastic neighbor embedding (tSNE) plot of object2vec from Fig. 1 according to the scores for each PC (Fig. 8 and Supplementary Fig. 7).

**Reporting summary**. Further information on research design is available in the Nature Research Reporting Summary linked to this article.

## Data availability
Stimuli, object2vec embeddings, spatial property ratings, and preprocessed fMRI data are available at the Open Science Framework repository for this project (https://osf.io/ug5zd/). The following publicly available resources were used in this work: ADE20K data set: https://groups.csail.mit.edu/vision/datasets/ADE20K/. Google News data set: https://code.google.com/archive/p/word2vec/. WordNet subset of Google News data set: https://github.com/chrisjmccormick/word2vec_matlab. AlexNet pre-trained on ImageNet: http://www.vlfeat.org/matconvnet/models/imagenet-caffe-ref.mat Source data are provided with this paper.

## Code availability
Code for the main analyses is available at the Open Science Framework repository for this project (https://osf.io/ug5zd/).

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

## Acknowledgements
We thank Caterina Magri for help with collecting behavioral ratings for the spatial properties model. We thank Rachel Metzgar for her help with fMRI data collection. This study was supported by NIH grant EY-022350 to RAE.

## Author contributions
M.F.B. and R.A.E. designed research; M.F.B. performed research; M.F.B. contributed analytic tools; M.F.B. analyzed data, and M.F.B. and R.A.E. wrote the paper.

## Competing interests
The authors declare no competing interests.
