## [Peer Review File · Nature Communications]

REVIEWER COMMENTS

Reviewer #1 (Remarks to the Author):

This paper provides an interesting demonstration of how machine learning tools capturing co-occurrence statistics of objects in scenes can be used to predict fMRI responses to individual objects in visual cortex. The method builds on earlier work using such models (e.g., from Gallant's group) to address the question of whether scene-selective cortex represents contextual associations between objects. The main finding is that responses in anterior PPA reflect object co-occurrence in scenes, suggesting a link between objects and their visual contexts. Another finding is that brain regions reflecting co-occurrences in scenes differ from brain regions reflecting co-occurrences in text, roughly mapping onto the scene vs object selectivity distinction.

General evaluation:

The study is well executed and the paper is a pleasure to read. The results are generally convincing (but see below). My main concern is that models of co-occurrence correlate with multiple dimensions, such that the results are interpretable in several alternative ways. As a result, it is not clear whether this study reveals new insight into object responses in scene-selective cortex.

Major comments:

-Relation with previous studies. Two studies (cited briefly in the Discussion) may deserve more extensive treatment in the Introduction. 1) The Stansbury et al. study had broadly similar objectives and also used similar methods, namely to "test the hypothesis that the human visual system represents scene categories that capture the statistical relationships between objects in the natural world. To investigate this issue, we used a statistical learning algorithm originally developed to model large text corpora to learn scene categories that capture the co-occurrence statistics of objects found in a large collection of natural scenes." And 2) the Sadeghi et al. study in which a similar object co-occurrence model was created and compared with a text-based similarity model. This study provided interesting information about the properties captured in these models that would seem relevant for the current study.

-Interpretability of the results. It is clear how the object2vec is created but it is not so clear what dimensions it captures. Objects that co-occur tend to share functional, semantic, and perceptual properties (Sadeghi et al. 2015). Some of these properties are thought to be encoded in the brain regions investigated here (e.g. shape in LO and pFs, size and portability in PPA). It thus remains plausible that the results reflect these properties independently of co-occurrence statistics. For example, previous work has shown that the PPA responds to large/stable objects (perhaps because of their landmark suitability or because they evoke a sense of space) and these objects may well co-occur (in outdoor scenes). To find out, one would need to dissociate these various dimensions experimentally (or add control models). Another possible explanation for the results is that the objects activated the associated scene representation (e.g. indoor/closed vs outdoor/open), and that the results thus reflect this scene-representation directly (e.g. spatial layout). I found that the paper does not seriously engage with these various explanations and remains rather vague in what exactly the results reveal. For example, there is

no effort to understand the object2vec model in relation to existing hypotheses regarding PPA selectivity. It seems like this could be done quite easily by relating, using RSA, the object2vec model with other models (e.g. using online ratings). These other models could also be directly related to the fMRI data.

-Sample size. Only four participants were tested, preventing population-level statistical inference. With so few participants it would be best to show the results of the individual participants (e.g., the brain maps and prediction accuracies) in the main text so that the reader can judge the generalizability and replicability. If not, it remains possible that results are driven by a single participant.

-Specificity of results. With only four participants it may not be so straightforward to run statistical tests. Still, the key dissociation in the paper – between object2vec and word2vec – would require more statistical support, for example in the form of direct contrasts in the whole-brain maps and direct comparisons within the ROIs (currently only the interaction with ant/pos PPA is presented). Similarly, the regional specificity is currently not strongly supported. The authors only note that object2vec was “numerically strongest in the PPA”.

Minor comments:

-What do you make of the lateralization in the whole-brain map? Was this consistent across participants?

-A previous study related activity patterns to objects from similar vs different scene contexts to each other in LO and PPA (MacEvoy & Epstein, 2011). How would you consider those results in relation to the current project?

-Regression line in Fig 7 looks off? In any case, it may be more informative to replace this graph with the individual subject data. Were any statistics performed on this analysis?

-I was a bit surprised by the definition of objects, which included scene elements such as rocks, cliffs, mountain, sea, and roads. How might this have influenced results?

-Fig 8: remove black background for easier printing (make it similar to Fig 1B)?

Reviewer #2 (Remarks to the Author):

Objects do not occur randomly in scenes, but rather have a rich covariance structure with scenes. In vision science, there is a long-standing idea that these contextual relationships are exploited by the brain to be able to efficiently understand the complexity of the visual world. Testing this hypothesis has been more difficult because these statistical relationships need to be measured, and it is difficult to divorce contextual relationships from the low-level features of the objects creating this context.

This work makes major contributions to solving both of these issues by leveraging advances in machine vision. Specifically, the measurement of object context was enabled through the use of a densely labeled scene database, and the authors appropriated methods from computational linguistics (word

embedding models) to create a model of object-scene contextual relationships (object2vec). Using voxelwise encoding methods, the authors found that scene-object context information was available primarily in the anterior portion of the PPA. Interestingly, comparing this to a word2vec model (that models words in context, and is not explicitly visual), the authors found partial overlap but with word2vec better explaining representations in early- and object-selective cortical regions.

This is impactful and creative work. There is considerable attention to detail here that is just delightful. Specifically, the way that backgrounds were created using DeepDreams in order to control for low-level visual similarity was very clever, and the use of a cross-validation procedure that forced their model to generalize across object categories ensures the robustness of the reported results. Overall, I have a few specific comments, but am otherwise happy to accept this paper.

-1- The automaticity of scene-object context

Throughout the paper, the authors write about how object context is “automatically” engaged. I feel like this reaches slightly beyond that can be concluded here. On page 22, the authors write that the task (warping detection) “required subjects to attend to each stimulus but did not require them to think about the mnemonic associations of the objects”. However, the presentation time of each object was fairly long (500 ms), and from looking at the warped images in the Supplementary Materials, it seems as though the task was likely fairly easy. Although it’s true that the task did not explicitly ask participants to think about context, it did not prevent them from thinking about context. In order to make claims about the automaticity, it seems that one would have to show that this contextual knowledge is activated even when doing so comes at a cost to task performance (similar logic to Stroop task). Therefore, I would encourage the authors to tone down this claim.

-2- Using AlexNet features as nuisance regressor

The AlexNet nuisance regressor method is interesting but I am not sure what to make of it. Because there are very few exemplars per category (10), and because the feature vectors in the convolutional layers of AlexNet are large and sparse, it’s not clear to me what one would obtain by averaging across these exemplars. It seems likely that one might not remain on an object manifold at all. Taking the principal components of this large representation may help, but I am left wondering how much variability these nuisance regressors explained. If they do not explain much variability, then the authors might want to look for an alternative method of controlling for low-level feature differences.

-3- Similarities between object2vec and word2vec

How similar were the object2vec and word2vec embeddings to one another? I imagine that the statistical regularities captured by object2vec may be a subset of the word2vec information (that also contains relationships with verbs, non-visual adjectives, etc). It seems that object2vec and word2vec were used in two separate encoding analyses. To what extent does the partial overlap observed in the searchlight analysis correspond to the variability shared between these two models? The authors may wish to consider using object2vec and word2vec together to use word2vec as a nuisance regressor.

-4- Format of the contextual representations

There is a lot that is surprising to me about the word2vec results in particular, and the authors do not spend a good deal of time discussing it or its implications. Specifically, it seems like word2vec might be capturing more multi-modal contextual representations because local word context can capture non-visual aspects of objects (sounds, smells, etc) as well as affordances related to the object in the form of verbs. From this perspective, it's pretty surprising that this model has some explanatory power for EVC. What are the authors' thoughts on this issue?

-5- Image category selection

I am curious about how the ten object classes were chosen. Some seem like they would have very rich and specific object context (e.g. toaster), while others would have rich but varied context (e.g. mugs that could occur in a variety of indoor spaces), while others seem to have very sparse object context (e.g. airplanes tend to have little else but sky or tarmac). If anything, this does not discount the findings of this paper because this paper shows that the method can generalize across these differences, but I am personally curious about the selection.

Small issues

Figure 8 is very confusing because the PC arrangement that the authors intend to show is shown in color on top of a spatial arrangement from two previous t-SNE dimensions.

For cleaning up ADE-20k labels, it would be helpful to have a list of changes. This will aid in the reproducibility of your results when others wish to use this method. Additionally, it is not clear whether the 3,148 unique labels are the raw unique labels, or the final labels after this procedure has been completed.

It seems as though there were different dimensionality criteria applied to the object2vec and word2vec models. While a threshold criterion was applied to object2vec (90% of variability in first 8 PC dimensions), a scree plot was applied to word2vec. I would like for the authors to comment on this choice and to provide a cumulative variability explained by the word2vec model in order to understand how comparable this is to object2vec.

Reviewer 1

This paper provides an interesting demonstration of how machine learning tools capturing co-occurrence statistics of objects in scenes can be used to predict fMRI responses to individual objects in visual cortex. The method builds on earlier work using such models (e.g., from Gallant's group) to address the question of whether scene-selective cortex represents contextual associations between objects. The main finding is that responses in anterior PPA reflect object co-occurrence in scenes, suggesting a link between objects and their visual contexts. Another finding is that brain regions reflecting co-occurrences in scenes differ from brain regions reflecting co-occurrences in text, roughly mapping onto the scene vs object selectivity distinction.

General evaluation:

The study is well executed and the paper is a pleasure to read. The results are generally convincing (but see below). My main concern is that models of co-occurrence correlate with multiple dimensions, such that the results are interpretable in several alternative ways. As a result, it is not clear whether this study reveals new insight into object responses in scene-selective cortex.

We thank for the reviewer for this positive evaluation.

Major comments:

-Relation with previous studies. Two studies (cited briefly in the Discussion) may deserve more extensive treatment in the Introduction. 1) The Stansbury et al. study had broadly similar objectives and also used similar methods, namely to "test the hypothesis that the human visual system represents scene categories that capture the statistical relationships between objects in the natural world. To investigate this issue, we used a statistical learning algorithm originally developed to model large text corpora to learn scene categories that capture the co-occurrence statistics of objects found in a large collection of natural scenes." And 2) the Sadeghi et al. study in which a similar object co-occurrence model was created and compared with a text-based similarity model. This study provided interesting information about the properties captured in these models that would seem relevant for the current study.

We agree, and we have added a paragraph to the introduction to address this related work.

-Interpretability of the results. It is clear how the object2vec is created but it is not so clear what dimensions it captures. Objects that co-occur tend to share functional, semantic, and perceptual properties (Sadeghi et al. 2015). Some of these properties are thought to be encoded in the brain regions investigated here (e.g. shape in LO and pFs, size and portability in PPA). It thus remains plausible that the results reflect these properties independently of co-occurrence statistics. For example, previous work has shown that the PPA responds to large/stable objects (perhaps because of their landmark suitability or because they evoke a sense of space) and these objects may well co-occur (in outdoor scenes). To find out, one would

need to dissociate these various dimensions experimentally (or add control models). Another possible explanation for the results is that the objects activated the associated scene representation (e.g. indoor/closed vs outdoor/open), and that the results thus reflect this scene-representation directly (e.g. spatial layout). I found that the paper does not seriously engage with these various explanations and remains rather vague in what exactly the results reveal. For example, there is no effort to understand the object2vec model in relation to existing hypotheses regarding PPA selectivity. It seems like this could be done quite easily by relating, using RSA, the object2vec model with other models (e.g. using online ratings). These other models could also be directly related to the fMRI data.

This is a good point. To address it, we have performed new analyses of three spatial properties of objects that previous work has associated with responses in scene regions: real-world size, spatial stability, and spatial definition. First, we collected behavioral ratings of these properties for each object in our experiment. We then ran encoding models using these spatial-property ratings as regressors, and we directly compared the results of this model with those from object2vec. As expected, we found that object2vec and the spatial properties model explained variance in overlapping portions of the PPA and the other scene-selective ROIs. However, we also found that object2vec explains additional variance in anterior PPA and parahippocampal cortex. These new data and results have been incorporated into the main text.

-Sample size. Only four participants were tested, preventing population-level statistical inference. With so few participants it would be best to show the results of the individual participants (e.g., the brain maps and prediction accuracies) in the main text so that the reader can judge the generalizability and replicability. If not, it remains possible that results are driven by a single participant.

We agree that single-subject results should be included in this manuscript. However, there are already a large number of figures in the results section, and showing all the single-subject plots would make the results section overly complex and dense. It is common practice in other papers using similar encoding-model methods to show the results from a single representative subject in the main text and to include the results from all subjects in the supplementary materials (e.g., work from the Gallant lab). Here we took the approach of showing the group-average result in the main text rather than a single representative subject. We have now also added single-subject results for all relevant analyses in the supplementary materials. Note that because we now report single-subject whole-brain results, we have switched to using FDR correction instead of the more stringent FWE correction. FDR correction is commonly used in other work reporting single-subject voxelwise encoding models (e.g., Huth et al., 2016; Lescroart & Gallant, 2019).

-Specificity of results. With only four participants it may not be so straightforward to run statistical tests. Still, the key dissociation in the paper – between object2vec and word2vec –

would require more statistical support, for example in the form of direct contrasts in the whole-brain maps and direct comparisons within the ROIs (currently only the interaction with ant/pos PPA is presented). Similarly, the regional specificity is currently not strongly supported. The authors only note that object2vec was “numerically strongest in the PPA”.

We have now added a whole-brain direct comparison of object2vec and word2vec, which shows significantly higher prediction accuracy for object2vec in clusters in anterior PPA (Fig. 7). In contrast, word2vec has significantly higher prediction accuracy in more lateral visual regions, around the areas of pFs and LO.

Minor comments:

-What do you make of the lateralization in the whole-brain map? Was this consistent across participants?

We did observe a trend for stronger object2vec results in the right hemisphere, but this does not appear to be consistent in all subjects, and we have not drawn any conclusions from it.

-A previous study related activity patterns to objects from similar vs different scene contexts to each other in LO and PPA (MacEvoy & Epstein, 2011). How would you consider those results in relation to the current project?

The MacEvoy study used multivoxel pattern analysis and looked for evidence of greater pattern similarity between objects from the same context vs. different contexts. They found no consistent evidence for the representation of contextual similarity in PPA, LO, and pFs (note that one out of their three experiments did show an effect in LO). Although we do not know for certain why the MacEvoy study did not find evidence of contextual representations, there are a number of aspects of the study that may have made it suboptimal for detecting such effects. First, the study only examined 8 object categories (2 each from 4 contexts), whereas here we examine 81 categories. Second, the more basic pattern classification methods used in MacEvoy may have been less sensitive than the encoding model approach used here. Finally, MacEvoy examined multivoxel patterns across the entire PPA rather than splitting the PPA into anterior and posterior segments. Our voxelwise modeling procedure shows that the object2vec encoding models have greater prediction accuracy in anterior PPA and parahippocampal cortex than in posterior PPA. We have added a new comment on the MacEvoy study in the Discussion section.

-Regression line in Fig 7 looks off? In any case, it may be more informative to replace this graph with the individual subject data. Were any statistics performed on this analysis?

We checked, and the lines in these plots are correct. Perhaps it is hard to judge by eye because the overlapping plot markers may not fully convey the density of points at different regions of the scatter plots. We have replaced this figure with the single-

subject plots. We consider this an exploratory analysis, which we developed to quantify an unexpected trend that we observed in our data. We have therefore not attempted to assign a p-value to these correlations or to draw any strong conclusions from them. Furthermore, it is not clear how to handle the spatial autocorrelation of voxels (which makes them dependent observations) when trying to calculate a p-value for these correlations.

-I was a bit surprised by the definition of objects, which included scene elements such as rocks, cliffs, mountain, sea, and roads. How might this have influenced results?

The stimuli for this experiment were guided by the labeled segmentation classes in ADE20K, which included not only independent objects but also fixed scene elements, like mountains and roads. We considered using alternative terms to describe these items (e.g., objects and stuff). However, there does not appear to be a generally accepted and detailed taxonomy for these concepts in the field, and it was unclear to us what criteria one should use to discriminate between objects and other potential classes of scene elements (e.g., Is a tree an object or something else?). We therefore decided to use the term “objects” here to refer to the scene elements that one would identify if asked to name the things in a picture (as in the labeling procedure for ADE20K). A benefit of this approach is that it is unbiased in the selection of nameable scene elements. An alternative approach of restricting the study to independent objects may run the risk of not generalizing to the much richer set of scene elements that are important to contextual processing in natural visual experience. It is also worth noting that previous studies of contextual associations included similar fixed scene elements as stimuli. For example, the Bar & Aminoff 2003 study on contextual associations included objects such as barn, roulette table, and windmill, and the Stansbury et al. 2013 study on object co-occurrence included objects such as sky, sand, and floor. We have added a discussion of this issue to the Methods section.

-Fig 8: remove black background for easier printing (make it similar to Fig 1B)?

We have made the background white for these plots.

Reviewer 2

Objects do not occur randomly in scenes, but rather have a rich covariance structure with scenes. In vision science, there is a long-standing idea that these contextual relationships are exploited by the brain to be able to efficiently understand the complexity of the visual world. Testing this hypothesis has been more difficult because these statistical relationships need to be measured, and it is difficult to divorce contextual relationships from the low-level features of the objects creating this context.

This work makes major contributions to solving both of these issues by leveraging advances in machine vision. Specifically, the measurement of object context was enabled through the use of a densely labeled scene database, and the authors appropriated methods from computational linguistics (word embedding models) to create a model of object-scene contextual relationships (object2vec). Using voxelwise encoding methods, the authors found that scene-object context information was available primarily in the anterior portion of the PPA. Interestingly, comparing this to a word2vec model (that models words in context, and is not explicitly visual), the authors found partial overlap but with word2vec better explaining representations in early- and object-selective cortical regions.

This is impactful and creative work. There is considerable attention to detail here that is just delightful. Specifically, the way that backgrounds were created using DeepDreams in order to control for low-level visual similarity was very clever, and the use of a cross-validation procedure that forced their model to generalize across object categories ensures the robustness of the reported results. Overall, I have a few specific comments, but am otherwise happy to accept this paper.

We thank the reviewer for this positive assessment.

-1- The automaticity of scene-object context

Throughout the paper, the authors write about how object context is “automatically” engaged. I feel like this reaches slightly beyond that can be concluded here. On page 22, the authors write that the task (warping detection) “required subjects to attend to each stimulus but did not require them to think about the mnemonic associations of the objects”. However, the presentation time of each object was fairly long (500 ms), and from looking at the warped images in the Supplementary Materials, it seems as though the task was likely fairly easy. Although it’s true that the task did not explicitly ask participants to think about context, it did not prevent them from thinking about context. In order to make claims about the automaticity, it seems that one would have to show that this contextual knowledge is activated even when doing so comes at a cost to task performance (similar logic to Stroop task). Therefore, I would encourage the authors to tone down this claim.

We agree, and we believe that the notion of automaticity is not important to any of our theoretical claims. We have therefore removed any mention of automaticity from the manuscript.

-2- Using AlexNet features as nuisance regressor

The AlexNet nuisance regressor method is interesting but I am not sure what to make of it. Because there are very few exemplars per category (10), and because the feature vectors in the convolutional layers of AlexNet are large and sparse, it's not clear to me what one would obtain by averaging across these exemplars. It seems likely that one might not remain on an object manifold at all. Taking the principal components of this large representation may help, but I am left wondering how much variability these nuisance regressors explained. If they do not explain much variability, then the authors might want to look for an alternative method of controlling for low-level feature differences.

This is a good question. To address this, we ran voxelwise encoding models using only the AlexNet PCs as regressors (i.e., without obj2vec or word2vec regressors) and we included the results from this analysis in the supplementary materials (Fig. S10). We found that the AlexNet PCs indeed explained a large amount of variance throughout much of visual cortex.

-3- Similarities between object2vec and word2vec

How similar were the object2vec and word2vec embeddings to one another? I imagine that the statistical regularities captured by object2vec may be a subset of the word2vec information (that also contains relationships with verbs, non-visual adjectives, etc). It seems that object2vec and word2vec were used in two separate encoding analyses. To what extent does the partial overlap observed in the searchlight analysis correspond to the variability shared between these two models? The authors may wish to consider using object2vec and word2vec together to use word2vec as a nuisance regressor.

The word2vec and object2vec models do indeed have overlapping explained variance in the whole-brain plots. We report a new analysis in this revision in which we directly compared the whole-brain effects of these models (Fig. 7). The results show that although object2vec and word2vec each have clusters of significantly higher prediction accuracy, there are also many voxels in which the models do not differ (as evidenced by the significant voxels in Fig. 6 that are not significant in Fig. 7). We believe that the results in Figures 6 and 7 convey the degree to which these models have overlapping and differential effects.

-4- Format of the contextual representations

There is a lot that is surprising to me about the word2vec results in particular, and the authors do not spend a good deal of time discussing it or its implications. Specifically, it seems like word2vec might be capturing more multi-modal contextual representations because local word context can capture non-visual aspects of objects (sounds, smells, etc) as well as affordances related to the object in the form of verbs. From this perspective, it's pretty surprising that this model has some explanatory power for EVC. What are the authors' thoughts on this issue?

This is a good point. We were also surprised by some of the anatomic effects of word2vec. Because these results were not hypothesized, we have been cautious in drawing conclusions from them. Nonetheless, we believe they are intriguing and worth further exploration. We speculated about two possible explanations in the Discussion (quoted below). The second explanation might account for the EVC effects. It's also worth noting that previous studies examining abstract semantic models of visual objects often report effects in low-level visual regions (e.g., Clarke & Tyler, 2014).

“Thus, one possibility is that the regions predicted by word2vec and object2vec encode different domains of abstract semantic associations that are not perceptual in nature. Some previous work that is consistent with this more semantically oriented interpretation has shown that the posterior fusiform gyrus and parahippocampal cortex are involved in the processing of object semantics, even when the experimental stimuli are words rather than images (but note that these studies did not specifically compare taxonomic and thematic associations)^{44,45}. However, a different, but not mutually exclusive, possibility follows from the fact that objects from similar taxonomic categories tend to have similar shapes. Insofar as the word2vec embeddings for object terms in language capture information about object shape, they may be better predictors of perceptual responses in object-selective visual cortex.”

-5- Image category selection

I am curious about how the ten object classes were chosen. Some seem like they would have very rich and specific object context (e.g. toaster), while others would have rich but varied context (e.g. mugs that could occur in a variety of indoor spaces), while others seem to have very sparse object context (e.g. airplanes tend to have little else but sky or tarmac). If anything, this does not discount the findings of this paper because this paper shows that the method can generalize across these differences, but I am personally curious about the selection.

We did not take into account our intuitions about object context when selecting the 81 object categories. We considered these issues at the start of the project, but we hypothesized that our objective modeling approach should be able to handle arbitrary contextual information and should thus not require constraints based on our subjective intuitions. The only constraints on stimulus selection were that the label for the object in ADE20K also needed to be present in the vocabulary for word2vec and that we were able to find high-quality images from which the objects could be isolated. The most limiting factor was how readily we could obtain high-quality images in which the object was either already isolated or could easily be isolated using Photoshop. Thus, the object categories were chosen based on two factors: 1) whether they were present in ADE20K and the word2vec vocabulary and 2) whether we could readily find high-quality images in our image search.

Small issues

Figure 8 is very confusing because the PC arrangement that the authors intend to show is shown in color on top of a spatial arrangement from two previous t-SNE dimensions.

We have added some additional explanation of these plots in the legend to clarify the difference between the colors and the spatial positions and to explain the logic of plotting the PCs in this way. To see if there was any additional way to address the reviewer's point, we did explore other ways of plotting these PCs, but ultimately we did not find any of them satisfactory, so we decided to stick with the original plots (with the exception of changing the background from black to white). Compared to possible alternatives, the current plots provided the easiest comparison of the PCs with the original embeddings before they were projected onto the encoding-model PCs, and the easiest comparison between PCs.

For cleaning up ADE-20k labels, it would be helpful to have a list of changes. This will aid in the reproducibility of your results when others wish to use this method. Additionally, it is not clear whether the 3,148 unique labels are the raw unique labels, or the final labels after this procedure has been completed.

Other than converting the labels into single-word labels (as described in the Methods), the other component of data cleaning was merging the annotations data from multiple possible labels that we identified as being associated with our 81 experimental object categories. We have now included a CSV file in the supplementary materials that indicates all object labels from ADE20K that were used for our 81 object categories. We also added a note in the Methods to specify that after this procedure, we were left with 2,959 unique object labels.

It seems as though there were different dimensionality criteria applied to the object2vec and word2vec models. While a threshold criterion was applied to object2vec (90% of variability in first 8 PC dimensions), a scree plot was applied to word2vec. I would like for the authors to comment on this choice and to provide a cumulative variability explained by the word2vec model in order to understand how comparable this is to object2vec.

We had different goals in mind when applying PCA to the object2vec and word2vec data. For object2vec, the model was already low-dimensional to begin with (by design), and PCA was used after combining the representations across many random instantiations of object2vec. Our goal was to retain all meaningful and orthogonal dimensions from these different random instantiations. In contrast, the goal of applying PCA to word2vec was explicitly to reduce the number of dimensions and avoid overfitting by our encoding models. Indeed, we observed that when retaining more PCs for word2vec (e.g., more than 50) the performance of our voxelwise encoding models generally became worse. We have added the suggested cumulative explained variance plot to the supplementary materials, but it is worth noting that it is not straightforward to compare the cumulative explained variance of the object2vec and word2vec PCs. The word2vec PCs were derived from high-dimensional

embeddings (i.e., 300 dimensions) and a dataset of 207,147 words, whereas the object2vec PCs were derived from low-dimensional embeddings (i.e., 10 dimensions) and a dataset of 2,959 objects. We have added notes to the Methods section that explain these issues.

REVIEWER COMMENTS

Reviewer #1 (Remarks to the Author):

The authors have addressed all my concerns.

Reviewer #2 (Remarks to the Author):

I thank the authors for their revision. Although my original concerns were adequately addressed, I do have a clarifying question about the new spatial properties analysis. In my understanding, a space defining object is one that significantly defines the shape of a space. A kitchen island is space defining because it changes the possible paths that can be traversed in the space, but an appliance like a blender would not be. Therefore, I am confused by the instructions given to the participants: "The answer choices were space defining, spatially ambiguous, or not classifiable, which were assigned to values 1 (space defining), 0 (spatially ambiguous), or 0.5 (not classifiable)." Why is there no option for unambiguously not space defining? What is an example of an object that is not classifiable and why? Further, I am concerned about what might happen to these 0.5 responses when averaging across the participants. More broadly, although the filtering procedure for all three spatial properties is sensible, I would like to see the level of agreement across participants in the three experiments. It seems that real-world size and mobility might have been much easier for participants to understand than spatial definition.

Reviewer 1

The authors have addressed all my concerns.

We thank for the reviewer for their positive evaluation.

Reviewer 2

I thank the authors for their revision. Although my original concerns were adequately addressed, I do have a clarifying question about the new spatial properties analysis. In my understanding, a space defining object is one that significantly defines the shape of a space. A kitchen island is space defining because it changes the possible paths that can be traversed in the space, but an appliance like a blender would not be. Therefore, I am confused by the instructions given to the participants: "The answer choices were space defining, spatially ambiguous, or not classifiable, which were assigned to values 1 (space defining), 0 (spatially ambiguous), or 0.5 (not classifiable)." Why is there no option for unambiguously not space defining? What is an example of an object that is not classifiable and why? Further, I am concerned about what might happen to these 0.5 responses when averaging across the participants. More broadly, although the filtering procedure for all three spatial properties is sensible, I would like to see the level of agreement across participants in the three experiments. It seems that real-world size and mobility might have been much easier for participants to understand than spatial definition.

For spatial definition, we used the same rating labels and corresponding values as in previous reports (Julian et al., 2016; Mullally & Maguire, 2011; Troiani et al., 2014). However, this reviewer comment led us to examine the split-half reliability of spatial-definition ratings, and we found that it was low ($r=0.08$). Because of this, we have decided to remove spatial-definition ratings from the spatial-properties encoding model. We found that our results were basically the same after removing spatial definition. Thus, real-world size and spatial stability are sufficient to account for the effects of the spatial-properties model. We also now report the split-half reliability of real-world size and spatial stability in the Methods section ($r = 0.99$ for real-world size and $r=0.94$ for spatial stability).

Julian, J. B., Ryan, J., & Epstein, R. A. (2016). Coding of Object Size and Object Category in Human Visual Cortex. *Cerebral Cortex*

Mullally, S. L., & Maguire, E. A. (2011). A New Role for the Parahippocampal Cortex in Representing Space. *Journal of Neuroscience*, 31(20), 7441–7449.

Troiani, V., Stigliani, A., Smith, M. E., & Epstein, R. A. (2014). Multiple Object Properties Drive Scene-Selective Regions. *Cerebral Cortex*, 24(4), 883–897.

REVIEWERS' COMMENTS

Reviewer #2 (Remarks to the Author):

I thank the authors for pointing me to the relevant references on the question and measures. They have addressed all of my concerns and I am happy to recommend publication.

REVIEWERS' COMMENTS

Reviewer #2 (Remarks to the Author):

I thank the authors for pointing me to the relevant references on the question and measures. They have addressed all of my concerns and I am happy to recommend publication.

We thank the reviewer for this positive assessment.